# SorryDB: Can AI Provers Complete Real-World Lean Theorems?

**Austin Letson** [1] **Leopoldo Sarra** [1] **Auguste Poiroux** [2 3] **Oliver Dressler** **Paul Lezeau** [4 5] **Dhyan Aranha** [6 7] **Frederick Pu** [8] **Aaron Hill** **Miguel Corredera Hidalgo** [9] **Julian Berman** [10] **George Tsoukalas** [11] **Lenny Taelman** [6]

## Abstract

We present SorryDB, a dynamically-updating benchmark of open Lean tasks drawn from 78 *real world* formalization projects on GitHub. Unlike existing static benchmarks, often composed of competition problems, hillclimbing the SorryDB benchmark will yield tools that are aligned with community needs, more usable by mathematicians, and more capable of understanding complex dependencies. Moreover, by providing a continuously updated stream of tasks, SorryDB mitigates test-set contamination and offers a robust metric for an agent's ability to contribute to novel formal mathematics projects. We evaluate a collection of approaches, including generalist large language models, agentic approaches, and specialized symbolic provers, over a selected snapshot of 1000 tasks from SorryDB. We show that current approaches are complementary: even though an agentic approach based on Gemini Flash is the most performant, it is not strictly better than other off-the-shelf large language models, specialized provers, or even a curated list of Lean tactics.

## 1. Introduction

Formal methods recast theorem proving as the computational problem of synthesizing machine-verifiable proofs (Moura & Ullrich, 2021). Subsequently, a trusted kernel can automatically verify a formal solution, and its error trace can be used to detect loopholes and hallucinations in reasoning. Recently, formal methods have gained renewed popularity in

---

[1]Axiomatic AI [2]Math, Inc. [3]EPFL [4]Imperial College [5]The London School of Geometry and Number Theory [6]University of Amsterdam [7]Côte d'Azur University [8]University of Toronto [9]ENSEIRB-MATMECA, INP-Bordeaux [10]Columbia University [11]The University of Texas at Austin. Correspondence to: Austin Letson <austin@axiomatic-ai.com>, Leopoldo Sarra <leopoldo@axiomatic-ai.com>.

*Proceedings of the $43^{rd}$ International Conference on Machine Learning*, Seoul, South Korea. PMLR 306, 2026. Copyright 2026 by the author(s).

mathematics. In particular, large-scale formalization efforts using the Lean interactive theorem prover (Moura & Ullrich, 2021), such as the Liquid Tensor Experiment (Commelin & Topaz, 2023), Carleson's theorem (Becker et al., 2025) and Fermat's Last Theorem (Buzzard & Taylor, 2026), have shown that systematically formalizing complete cutting-edge mathematical results is becoming feasible. Formalization improves collaboration among mathematicians, allowing them to unambiguously specify concepts of interest, reuse previously proven results, detect errors, and automatically validate a mathematical argument by checking its formalization, which for complex arguments can even decrease the costs of peer review (Hales, 2024; Robertson et al., 1997). Crucially, this work is rarely linear: mathematicians "prove in the open", with dozens of contributors working asynchronously across a shared codebase.

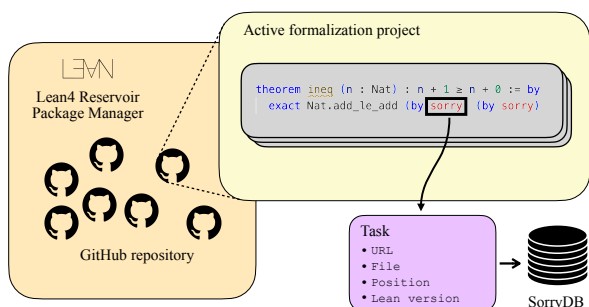

*Figure 1.* Dataset extraction pipeline for SorryDB. We consider 78 repositories of active GitHub Lean projects and list all the theorems that contain a sorry. We save the metadata for each of them and store it in the database. A project is considered active when it had at least one commit in the previous three months.

In recent years, formal methods have also been adopted by researchers focusing on automated mathematical reasoning. Outstanding results have been shown, for example, by achieving a performance that would deserve medals at the International Mathematical Olympiad (Hubert et al., 2025; Achim et al., 2025; Chen et al., 2025b). Rather than only proposing the solution to a problem (Huang & Yang, 2025), these models can also provide an automatically verifiable proof. This allows for post-selecting correct solutions (Lin

et al., 2025a), or even using the feedback as a strong reward signal for reinforcement learning training (Lambert et al., 2025) with verifiable rewards (Ren et al., 2025; Wang et al., 2025a).

State-of-the-art models are mainly being evaluated on competitive math datasets, for example, on IMO and other olympiad-level mathematics, such as miniF2F (Zheng et al., 2021) or undergraduate-level math challenges, such as PutnamBench (Tsoukalas et al., 2024). However, competitive math is not fully aligned with practical applications, and, since it does not cover actual projects, it may not be fully representative of the real challenges of formal theorem proving, including a broader mathematical context, the dependence on theorems and definitions defined in the specific project itself (Hu et al., 2025; Kumarappan et al., 2025), or simply the messiness that comes with fitting solutions into real-world applications. Moreover, these benchmarks are essentially saturated, and it becomes increasingly difficult to evaluate the performance of different automated theorem provers, thus hindering progress in the field (Reuel et al., 2024). In addition, as the datasets are made public and agentic systems using standard large language models (LLMs) are developed (Varambally et al., 2025; Ospanov et al., 2025; Wang et al., 2025c), it becomes challenging to detect solution leakage in training data and distinguish memorization from more general reasoning skills.

Active formalization projects offer an ideal test bed for addressing the limitations of current benchmarks. Starting from an informal list of definitions and theorems that lead to the main result of a project, developers first formalize the statements of theorems and their dependencies. Unproven theorems or parts of them are temporarily marked with a `sorry` placeholder. This process mostly happens in open repositories on GitHub, in order to enable large-scale collaboration, and provides a steady supply of novel challenges that Lean practitioners have left for others, or their future selves, to solve.

In this paper, we propose SorryDB, a new evaluation dataset based on real formalization projects. This dataset is specifically meant to evaluate the practical usefulness of an automated theorem prover by resembling the envisaged day-to-day usage. It also addresses the problem of saturation, misalignment, and leakage of existing benchmarks. More specifically, we aim to replicate the spirit of SWE-bench (Jimenez et al., 2024), meant for software engineering tasks, for theorem proving: evaluate state-of-the-art models on tasks as close as possible to real applications that are valuable to the community. Our main contributions are:

1. **The SorryDB Dataset**: We design a procedure to select the parts of a theorem that have not been proven yet (i.e. marked with a "sorry") from open-source Lean

repositories and assemble them into a dynamically-evolving dataset, by construction aligned to practical applications, as shown in Figure 1.

2. **Benchmarking Framework**: We provide an automated validation mechanism to evaluate whether a proposed solution is correct or not and allow assessing the performance of an automated prover on the SorryDB dataset.

3. **Evaluation**: We analyze a static snapshot of the dataset with 1000 tasks and a cutoff set to January 2026, and provide a baseline evaluation of a collection of models. We confirm that the proposed dataset provides useful feedback on the practical usefulness of automated theorem provers, that it is not saturated yet, and we analyze the most common failure cases. The evaluation highlights that current models are complementary, solving different subsets of problems, and that overall the iterative feedback is the dominant factor in performance, with self-correcting and agentic approaches outperforming one-shot pass@k approaches.

## 2. Background and Related Work

**Interactive Theorem Provers.** To create a formal proof for a mathematical result, the first step is to convert the natural language statement into a formal language. The formal statement will define precisely all the mathematical objects, include all the assumptions necessary to prove the theorem, and clearly specify the target goal. Subsequently, the proof is provided, either by modifying the goal step by step until it becomes a trivial consequence of the hypotheses or by directly instantiating an object with the properties required by the goal. Libraries of definitions and proven theorems, such as Mathlib (mathlib Community, 2020), help import the appropriate objects and build on top of existing results. In Lean, it is also possible to write only the formal definition of a theorem, putting a placeholder keyword, written as `sorry`, in place of the proof. In the context of formalization projects, `sorry` statements are often left as work items to be completed later in the project.

**Competition Math.** Most common datasets to evaluate automated theorem provers focus on competition math. The most popular are PutnamBench (Tsoukalas et al., 2024), focused on the Putnam competition, a popular annual contest for undergraduate math students, and miniF2F (Zheng et al., 2021), a curated collection of formalized olympiad-level math problems. DeepSeek ProverBench contains hundreds of formalized problems from textbooks, tutorials, and AIME competitions (Ren et al., 2025). Several other datasets exist, including ProofNet (Azerbayev et al., 2023) and IndiMath-Bench (Biyani et al., 2025).

Since these datasets are focused on competitive math, they cover only a small segment of the real-world use cases of formal methods. Moreover, they are now mostly saturated, and their solutions may have been leaked in the training data of many off-the-shelf models.

The reduced performance of current state-of-the-art Lean theorem provers on tasks outside competition math has been documented in various works, for example, as the motivation for building datasets focused on specific math fields, such as FATE, assessing the performance on frontier algebra (Jiang et al., 2025) and LeanCAT, on category theory (Xu et al., 2025).

**Formalization Projects.** FormalML (Yang et al., 2025c) evaluates fill-in-the-sorry tasks from real Lean projects, mainly on two selected projects on optimization and probability theory. RLMEval (Poiroux et al., 2025a) also evaluated neural theorem provers on six existing popular Lean formalization projects, focusing on testing the ability to prove again the main theorems of the formalization project. These are relatively small curated datasets on specific fields of mathematics. Along these lines, we propose to systematically build a dataset on a larger scale, explicitly meant for theorem prover evaluation that resembles the day-to-day usage of a theorem prover by a Lean developer.

Some other work has compiled a general dataset by scraping multiple repositories of Lean projects. In particular, miniCTX (Hu et al., 2025) sources theorems from real projects and formalized textbooks. The authors also emphasize that a real theorem proving task requires local context and may be much more difficult than proving a competition math theorem out of context. We propose to push this assumption even further and build a much larger, dynamic dataset that contains theorems whose solution has not yet even been published, mitigating data contamination risk, instead of trying to prove again already established theorems.

**Benchmarking Practical Utility.** The idea of using challenges from real applications is very popular for software engineering coding agents, e.g. SWE-bench (Jimenez et al., 2024; Yang et al., 2024; Wang et al., 2025b). In this case, the task is to fix a given issue taken from a real GitHub repository by submitting the equivalent of a pull request. Another example is CodeSearchNet (Husain et al., 2020) for the evaluation of semantic code search, containing millions of real-world functions associated with their documentation.

**Open-source provers.** Several models have been fine-tuned from general-purpose large language models for theorem proving. In particular, Kimina Prover (Wang et al., 2025a), Goedel Prover (Lin et al., 2025b) and DeepSeek Prover (Ren et al., 2025) employ reinforcement learning to learn to prove formally. Hilbert (Varambally et al., 2025)

uses an agentic decomposition harness that combines informal reasoning with formal proving without training, achieving strong performance in competition problems.

**Proprietary provers.** AlphaProof (Hubert et al., 2025) was the first model to use Lean to prove Olympiad-level problems using reinforcement-learning-driven search inspired by AlphaZero (Silver et al., 2017). Several theorem provers based on large language models have recently been announced, including SeedProver V1.5 (Chen et al., 2025a), Aristotle (Achim et al., 2025), or Axiom (Axiom, 2026). More recently, Logical Intelligence claims to have even saturated PutnamBench with a performance of $99.4\%$ (Logical Intelligence). These provers are often available in closed beta programs with usage terms that would not allow running systematic evaluations.

## 3. SorryDB

In this section, we detail SorryDB, which consists of both the dynamically updating benchmark and the framework that allows extraction and verification of solutions.

### 3.1. Dataset

We create SorryDB by extracting proof obligations, i.e. placeholder `sorry` keywords, from active open-source Lean projects. They may be full unproven theorems or open intermediate steps within a sketched proof.

We call this evaluation task *fill in the sorry*: provide the code which satisfies the proof obligation captured by the `sorry` keyword inside a theorem. In other words, in case a theorem contains multiple `sorry` statements, we do not require that one prove the entire theorem, but only the target proof obligation. This choice of granularity is deliberate: it reflects the actual workflow of Lean practitioners, who formalize theorem statements and proof sketches first, leaving specific obligations as `sorry` placeholders to be discharged later. Moreover, proving a full theorem is a special case of filling a sorry (when the entire proof body is a single `sorry`), so this task formulation is strictly more general.

As shown in Figure 1, the dataset generation procedure consists of the following phases: repository selection, where we select which projects to include; and extraction, where we extract and validate the `sorry` statements.

**Repository selection criteria.** We consider GitHub repositories from Reservoir, the Lean package registry (Lean Prover contributors, 2026). As a consequence, all included repositories satisfy its inclusion criteria (Lean FRO, 2025), and most notably are licensed under a GitHub-recognized OSI-approved license. In addition, we require that the project is still maintained by verifying that it has been up-

dated at least once since January 2025, and that its GitHub repository is public.

To obtain a large variety, SorryDB indexes `sorry` statements from a diverse set of 78 repositories (listed in full in Appendix E) which broadly fall into the following hand-crafted categories:

- **Pedagogical:** Teaching materials, tutorials, course repositories, and learning resources.

- **Tooling:** Lean-specific tools, utilities, and development aids (such as *verso* or *duper* (Tang, 2025)).

- **Benchmark:** AI evaluation datasets, benchmarks, and math competition problems (such as *PutnamBench* & *miniF2F*).

- **Library:** Reusable mathematical or programming libraries (such as *batteries*, *cslib*, and *mathlib4* (mathlib Community, 2020)).

- **Formalization:** Projects formalizing specific theorems or mathematical theories (such as *FLT* (Buzzard & Taylor, 2026) or *Carleson* (Becker et al., 2025)).

These categories reflect the variety of Lean projects found in the wild: projects are typically created to teach, to provide developer tools, to collect competition problems, to serve as reusable libraries, or to formalize a specific mathematical result. Since a repository's purpose is typically fixed at creation, each repository receives a single permanent category label. Including all five categories means the dataset covers the full range of Lean proof obligations in practice, from simple pedagogical exercises to open research problems.

**Task extraction.** For each selected repository, the indexer extracts all `sorry` keywords and stores identifying metadata, including tasks found on both main and work-in-progress branches. Subsequently, these candidates are filtered to ensure they can be reproduced. We then filter for tasks that strictly require a proof and deduplicate them to keep only the most recent occurrence. See Appendix D for full details of extraction, filtering and deduplication.

### 3.2. Benchmarking using SorryDB

We can use the SorryDB dataset for evaluating automated theorem provers. When evaluating a proposal, we can deterministically validate it by checking if the code compiles and if the `sorry` has been properly completed, without the need for a curated label. The evaluation metric is just the average success rate of a model, i.e. the accuracy.

We provide an automated validation pipeline for the evaluation of each proposal. To verify that the proposed solution for a given `sorry` keyword is correct, we replace the

`sorry` with the proposal and use LeanInteract (Poiroux et al., 2025b) to check the proof state at that specific location, and validate that it compiles successfully without any `sorry`-like term in the final state. Just building the Lean project is insufficient to verify that the sorry has been filled by the prover, because `sorry` is syntactically valid in Lean. In addition, there are alternative keywords or constructs that can have a similar effect to a `sorry` placeholder. Since the evaluation requires cloning and building the specific project repository for each `sorry`, which can be space-intensive and time-consuming, some engineering effort is required to run the dataset evaluation efficiently. Specifically, it is useful to prepare all the repositories in advance so that they can be quickly instantiated for verifying a specific proposal for the associated task.

We describe our implementation in more detail in Appendix F.

To evaluate a new prover on SorryDB, users can plug their solver into our verification pipeline using the framework released alongside the dataset. The most up-to-date instructions for running the benchmark and submitting solutions are maintained in the README of the SorryDB GitHub repository at https://github.com/SorryDB/SorryDB/.

## 4. Experiments

In our evaluation, we use `SorryDB-2601`, a snapshot of the SorryDB dataset created with the procedure described in Section 3 with cutoff date of January 2026 and is thus named 2601. This dataset contains 5663 sorry filling tasks from 78 repositories. In particular, for practicality, we focus on a split of 1000 `sorry` statements, chosen to be the most recent according to the commit timestamp, and with the constraint of maximizing the variety of repositories in the slice. The details of the dataset with the distribution of tasks per category are shown in Table 1.

*Table 1.* Distribution of `sorry` statements by repository category in the `SorryDB-2601` dataset snapshot and for the specific 1000 `sorry` statements we sampled for use in our experiments. For building the latter, we sampled the newest `sorry` statements from each repository to maximize diversity and recency.

| CATEGORY | SORRYDB-2601 | TEST SAMPLE |
|---|---|---|
| FORMALIZATION | 3,099 | 562 |
| PEDAGOGICAL | 1,037 | 216 |
| BENCHMARK | 797 | 48 |
| LIBRARY | 616 | 89 |
| TOOLING | 114 | 85 |
| TOTAL | 5,663 | 1,000 |

According to the needs, alternative choices are possible: a larger time frame could have been chosen, or even the full dataset could have been evaluated, obtaining a more precise

but also more expensive and time-consuming evaluation. See Appendix I for a detailed analysis of token usage and inference cost across the considered models. See Appendix G for the specific procedure we used to select this set.

We consider several approaches ranging from deterministic tactics to general-purpose large language models, specialized models and iterative or agentic architectures. For each class of methods, we show a few representative models.

It is important to notice that the fill-in-the-sorry task requires filling a specific part of a theorem, and not the full theorem itself. For this reason, it requires strong understanding of the local proof context in addition to the more general project context. For these reasons, some provers may not support this task because they were designed to complete only full theorems, making this task appear slightly out-of-distribution.

**Baseline tactics.** A tactic is a short command or instruction that tells Lean how to construct a proof step-by-step, rather than having to write out the complex logical terms manually. The simplest prover we consider is the `trivial` tactic, capable of solving extremely simple theorems where the proof is a trivial reformulation of the goal. We also consider other deterministic tactics such as `ring`, `linarith`, `norm_num`, and more advanced approaches such as `simp` or `grind` (de Moura, 2025). `grind` serves as a high-powered, general-purpose automation engine, using SMT-based techniques to solve complex problems. See Appendix H for a full analysis of the deterministic approaches.

**Foundation models.** We consider state-of-the-art general-purpose large language models such as Claude Opus 4.5 (Anthropic, 2025), Gemini Flash 3 (Comanici et al., 2025), GPT 5.2 (OpenAI, 2025). We also consider Qwen 3 (Yang et al., 2025a) as an example of an open-weight model. Foundation models are trained on very large amounts of data and are known for their flexibility and generalization capabilities outside of their specific training domain.

**Specialized models.** We also include models specifically trained for automated theorem solving, such as Kimina-prover (Wang et al., 2025a) and Goedel-Prover V2 (Lin et al., 2025b). We specifically select these two specialized provers because they have good performance on competitive math datasets, are open-weight, and are compatible with the fill-in-the-sorry task.

**Self-correcting and agentic approaches.** We consider self-correcting approaches in which we sample each model iteratively up to 16 times. For every proposal, a deterministic reviewer inserts the solution into the project and tests for compilation errors. In case of failure, it returns the build error message as feedback for the next iteration. In addition,

*Table 2.* Performance of various provers on a snapshot of 1000 tasks from `SorryDB-2601`. Tactics are deterministic and thus only evaluated once. General-purpose and specialized models are evaluated by sampling once (pass@1) or by checking if any of 32 candidates solve the task (pass@32). Self-correcting (SC) or Agentic approaches run for up to 16 iterations. The row labelled as "Combined" shows how many tasks were solved by at least one prover. Since it is the largest value, it highlights the complementarity of the considered provers.

| APPROACH | PASS@1 | PASS@32 |
|---|---|---|
| *Deterministic* | | |
| TRIVIAL | 2.1 % | N/A |
| TACTICS | 8.4 % | N/A |
| *General-purpose LLM* | | |
| GPT 5.2 | 6.2 % | 13.2 % |
| CLAUDE OPUS 4.5 | 7.8 % | 15.4 % |
| GEMINI FLASH 3 | 10.8 % | 20.5 % |
| GEMINI PRO 3 | 11.0 % | 20.5 % |
| QWEN 3 | 5.0 % | 8.1 % |
| *Specialized LLM* | | |
| KIMINA PROVER 8B | 1.8 % | 7.7 % |
| GOEDEL PROVER V2 32B | 2.7% | 11.3 % |
| *Iterative* | | |
| CLAUDE OPUS 4.5 (SC) | 27.1 % | - |
| GEMINI FLASH 3 (SC) | 27.9 % | - |
| GEMINI FLASH 3 (AGENTIC) | 30.3 % | - |
| COMBINED | 35.7 % | |

we implement a simple baseline agentic approach inspired by most popular implementations (Jiang et al., 2023; Varambally et al., 2025; Cao et al., 2025; Requena et al., 2026). In particular, it includes a Proposer, tasked to provide a solution for the given task, and a deterministic reviewer, like in the pure self-correcting mode. The Proposer has a ReAct-style (Yao et al., 2023) architecture with a library search tool, LeanSearch (Gao et al., 2025). The proposer uses Gemini Flash 3 with a dynamic thinking budget and 5 rounds of tool calls at each iteration.

## 4.1. Results

We evaluate the approaches based on language models without self-correction with pass@32, while self-correcting models can run up to 16 iterations. We do not evaluate the tactics multiple times, as their outcome is deterministic. Performance is shown in Table 2. By looking at the results, we immediately notice that the feedback from the build output, which, in case of errors, includes the state before the failure and the error message, is extremely useful for the models we tested. As emphasized by Figure 3, with just 16 model calls, iterative approaches strongly outperform any single-shot approach, even if considering pass@k with double the number of calls. In other words, when working with SorryDB, access to the project environment and having

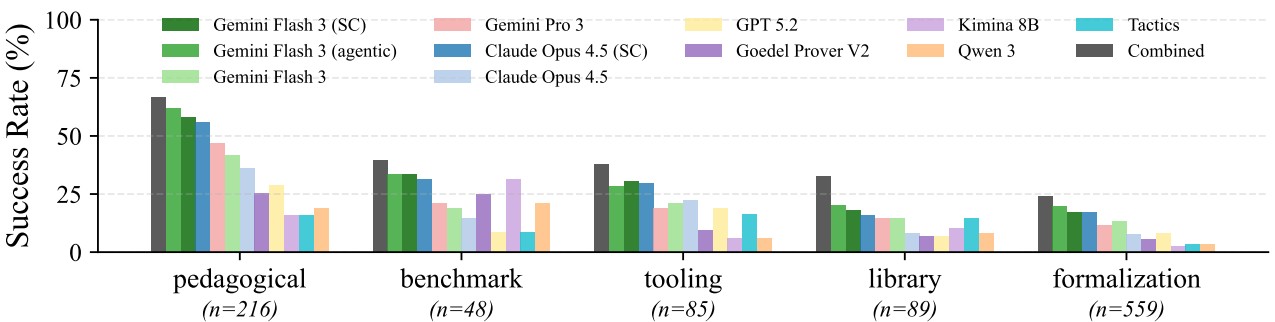

*Figure 2.* Success rate of different provers split by repository category. We compare general purpose LLMs, specialized models (pass@32), self-correcting (SC) and agentic approaches (16 iterations). We see that tasks from pedagogical repositories are easier to prove, while those from math formalization projects are harder. A specialized prover such as Goedel Prover works relatively better on benchmark repositories but has worse performance on other project types.

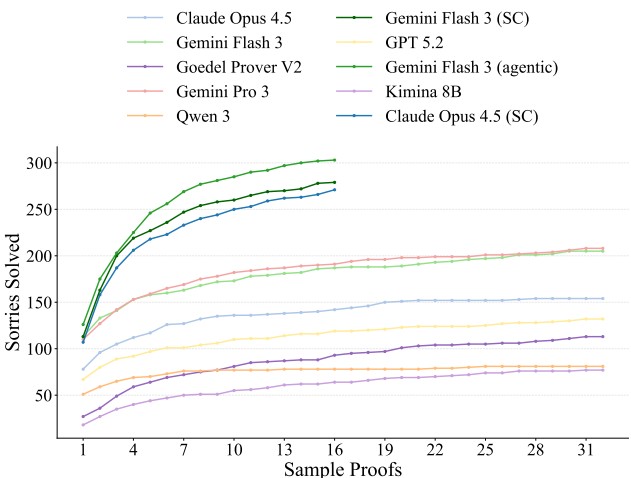

*Figure 3.* Number of solved tasks by number of LLM calls. Iterative sampling (self-correcting or agentic) is much more efficient than parallel sampling.

the ability to compile it to test a proposed solution is much more helpful than longer reasoning budget or fine-tuning on specific datasets.

**Performance depends on category.** Figure 2 shows that performance is generally higher in pedagogical repositories and repositories that contain existing benchmark problems. After all, pedagogical repositories often contain theorems that can be solved in a few steps, or even already exist as proven theorems in Mathlib, while benchmark problems are a common target for reasoning evaluation, and recent models may have been trained on them. Goedel Prover V2 and Kimina both perform relatively better on benchmark repositories.

**Different provers solve different tasks.** By inspecting the distribution of solved proofs per prover, we observe that aggregating solutions across provers increases the total

number of solved tasks. This indicates that provers are complementary: each succeeds on a different subset of problems, and even the strongest prover fails on instances that others can solve.

**Specialized provers underperform outside of competition math benchmarks.** The RL training for specialized models, Goedel Prover and Kimina, focused on competition math and this may explain the difficulty in generalizing to other domains. Furthermore, they have been trained on older Lean versions, and the version mismatch could be a reason for their weaker performance. Also, theorem provers specifically trained for solving entire theorems may find the fill-in-the-sorry task slightly out of domain.

### 4.2. Qualitative Analysis

A striking example from the Brownian-motion repository (Degenne et al., 2025) is shown in Figure 5. It involves proving that the Càdlàg property (right-continuous with left limits) is preserved under the squared norm operation. While standard LLMs were distracted by hunting for non-existent Mathlib lemmas, the self-correcting approach, without tools, succeeded by directly constructing the solution. This required a deep understanding of the local `IsCadlag` definition: the model manually initialized the structure's fields and successfully constructed an existential witness for the left limit. Interestingly, the tool-enabled agentic approach failed to solve this same problem, as it kept calling the search tool rather than constructing the proof term itself. On aggregate, the tool-enabled agent does not degrade performance and is in fact the strongest single configuration we evaluate (30.3% vs. 27.9% for the same model in self-correcting mode without tools). This specific example instead illustrates a more subtle trade-off: tool access can lead a generally stronger prover to over-rely on retrieval when the goal would have been better solved by directly unpacking a local definition. As shown in the UpSet plot

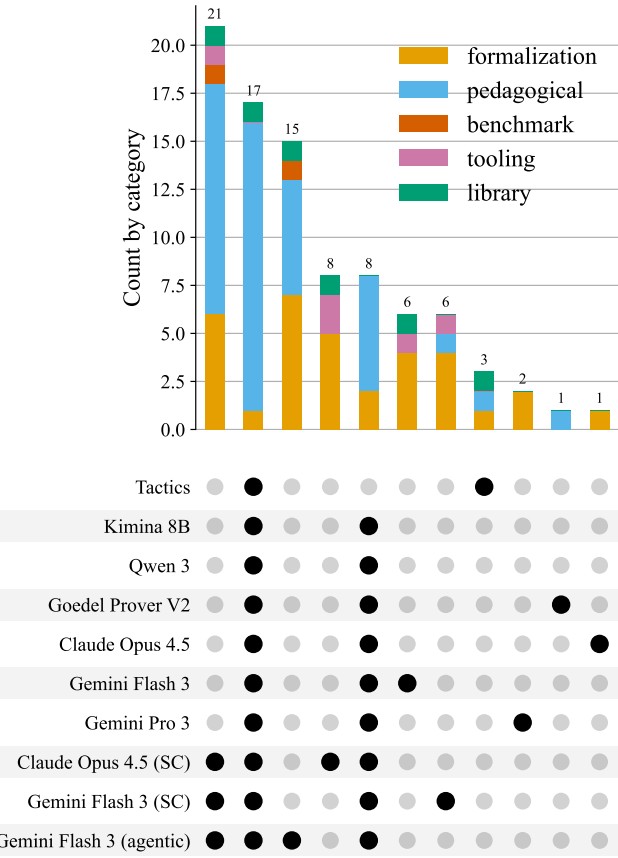

*Figure 4.* UpSet plot of `sorry` statements solved by each model. Each column represents the number of theorems solved by all the provers marked in the table with a black circle, and not solved by the others. Deterministic tactics solve some `sorry` statements that none of the LLM-based approaches could solve.

the existing library. It does not test reasoning power in a vacuum. By including tasks that demand deep library knowledge alongside those requiring novel construction, SorryDB provides a more holistic evaluation of the practical utility of an AI-based theorem prover.

## 5. Discussion

SorryDB indexes open proof obligations (i.e. `sorry` statements) in active Lean formalization projects, with an emphasis on projects that received commits within a recent timeframe. This choice is made to promote the selection of projects that are still under development. This design ensures that the benchmark keeps targeting new and unsolved tasks, rather than problems that are already well-supported by currently available proof automation. In practice, this means that each `sorry` represents a proof obligation that contributors have chosen not to discharge immediately, probably because it would require nontrivial insight, adding missing lemmas, or substantial refactoring. As a result, it is reasonable to expect that each task is, by construction, a meaningful unit of work for evaluating automated theorem provers. As automated theorem proving improves and these tools are used to solve `sorry` statements, the remaining ones naturally become more challenging, allowing SorryDB to adapt to the current state of the art and remain a moving benchmark that scales with progress rather than becoming saturated. In other words, this is an evolving dataset: while we analyze a static snapshot in our evaluations, we plan to take new snapshots of the dataset every six months. See Appendix J for the full snapshot versioning convention, release cadence, and contamination-control policy. As a consequence, SorryDB will not be saturated: as new theorem provers are released, mathematicians adopt them, and the difficulty of recent sorry-filling tasks will increase.

One drawback of this approach is not knowing the best possible performance on a snapshot: we measure the relative performance among provers, but we do not know how many tasks are actually unsolvable because they are unprovable. Trying to prove the negation of the goal for tasks without a known solution could be a way to estimate this in the future. Moreover, we could adopt in the future an Elo score (Király & Qian, 2017), such as that used in CodeClash (Yang et al., 2025b), to emphasize that the evaluation score is relative; see Appendix K for details and per-prover Elo ratings. As a future step, we plan to set up a leaderboard to compare the performance of the various state of the art models, and regularly provide an updated snapshot of the dataset.

**Limitations.** Regarding current limitations, there are a few directions open to future improvements.

First, despite the broader variety of repositories, the scope is still limited to projects in Lean Reservoir, and the data

(Figure 4), this is not an isolated incident as six theorems are solved by the self-correcting Gemini Flash but missed by the agentic version, and the reverse also happens. The two configurations are therefore complementary rather than strictly ordered.

This highlights the strength of SorryDB: it captures tasks that resist simple retrieval and require models to engage in novel arguments within a specific local context. A complementary success is observed in the RemyDegenne/CLT repository, where the task involved proving the integrability of a function on a product measure space. While pure generative models failed to reconstruct the complex measure-theoretic argument, both the deterministic tactics and the Agentic one successfully solved it by retrieving specific Mathlib lemmas (`integrable_comp_eval` and `integrable_comp_of_integrable`, respectively). This contrast highlights a key strength of SorryDB: it effectively captures the diverse reality of formalization, where success requires knowing when to construct a novel proof, as in the Brownian motion example, and when to leverage

**Incorrect Attempt - Goedel Prover:**

*The model outlines the proof structure but gives up on each step with* `sorry`.

```
have hX2_cadlag : ∀ ω, IsCadlag (fun t ↦ ‖X t ω‖ ^ 2) := by
  intro ω
  have h₁ : IsCadlag (X · ω) := hX_cadlag ω
  have h₂ : Continuous (fun x : E => ‖x‖ : E → ℝ) := by sorry
  have h₃ : IsRightContinuous (fun t : ι => ‖X t ω‖ ^ 2 : ι → ℝ) := by sorry
  have h₄ : IsLeftContinuous (fun t : ι => ‖X t ω‖ ^ 2 : ι → ℝ) := by sorry
  have h₅ : IsCadlag (fun t : ι => ‖X t ω‖ ^ 2 : ι → ℝ) := by sorry
  sorry
```

**Incorrect Attempt - Gemini (agentic with LeanSearch):**

*The model hallucinates a high-level lemma (*`continuous_comp`*) to fill the sorry.*

```
have hX2_cadlag : ∀ ω, IsCadlag (fun t ↦ ‖X t ω‖ ^ 2) :=
  fun ω ↦ (hX_cadlag ω).continuous_comp (continuous_norm.pow 2)
```

**Correct Solution - Gemini (self-correcting):**

*The model does not find an appropriate existing lemma and fills the sorry by constructing the proof term structure manually.*

```
have hX2_cadlag : ∀ ω, IsCadlag (fun t ↦ ‖X t ω‖ ^ 2) :=
  fun ω ↦ {
    right_continuous := fun t ↦
      ((hX_cadlag ω).right_continuous t).norm.pow 2
    left_limit := fun t ↦
      let ⟨l, hl⟩ := (hX_cadlag ω).left_limit t
      ⟨‖l‖ ^ 2, hl.norm.pow 2⟩
  }
```

*Figure 5.* Three attempts at filling the same `sorry` in the `brownian-motion` repository. (top) Goedel Prover outlines the proof structure but abandons each sub-goal with `sorry`, unable to find the required lemmas. (middle) Gemini with LeanSearch attempts a syntactically-correct guess using `continuous_comp`, a non-existent library function that it assumed to exist. (bottom) Gemini self-correcting (without LeanSearch) inspects the definition of `IsCadlag` and manually constructs a proof term by providing the two required fields: `right_continuous` and `left_limit`.

distribution may be biased towards some fields, while other topics may be underrepresented. Considering only projects in the Lean Reservoir restricts the eligible repositories, but it is more likely to lead to higher quality projects.

In addition, the verifier currently does not allow solutions that propose to add imports to the target file, open additional namespaces, or to define additional lemmas to use in the target theorem, which could be a limitation on the expressivity of the prover. This constraint can be easily removed in a future version of the benchmark.

Finally, some solutions may potentially exploit Lean properties to trick the verification pipeline. While we currently employ a custom script to remove instances of `sorryAx`, an exploit which the agentic systems used to get around the sorry verification, future work can easily improve the verification pipelines by integrating tools such as Leanchecker (leanprover contributors), SafeVerify (GasStationManager) or another custom kernel check tool.

Lastly, the evaluation goal of this paper was not to build the strongest possible agent, but rather to provide a simple

baseline and keep the focus on the dataset and evaluation protocol. We expect that more repository-aware agents will be able to challenge the baseline agent we provide, and we hope SorryDB will serve as a testbed for such future work.

## 6. Conclusion

In this work, we introduce SorryDB, a dynamic benchmark derived from unresolved `sorry` statements in Lean GitHub repositories containing active formalization efforts. By focusing on outstanding real-world tasks, SorryDB is inherently aligned with priorities of the formalization community. The benchmark is continually refreshed to mitigate data contamination and ensure it reflects the ever-evolving formalization landscape. Our evaluations demonstrate the efficacy of modern agentic provers while identifying specific limitations compared to traditional methods. With SorryDB, we aim to help herald the coming future of increasingly capable and usable proving assistants. Code is available at https://github.com/SorryDB/SorryDB/.

## Acknowledgements

We thank Morph Labs for providing support with their infrastructure, and Axiomatic AI for providing support with LLM inference. Part of this research was supported by the European Research Council (ERC), grant 864145. A.L. and L.S thank Marco del Tredici and Benjamin Breen for comments on the original manuscript and for discussions.

## Impact Statement

This paper presents work whose goal is to advance the field of Machine Learning. There are many potential societal consequences of our work, none which we feel must be specifically highlighted here.

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

# A. Dataset composition

The source for all GitHub repositories which store SorryDB `sorry`s is the Reservoir Lean package registry. This means that selected repositories comply with the Reservoir inclusion criteria and have a root `lake-manifest.json` file which specifies relevant descriptive metadata. All branches are included; when the indexer is run, it looks for "leaf commits," which are the final commits in any branch.

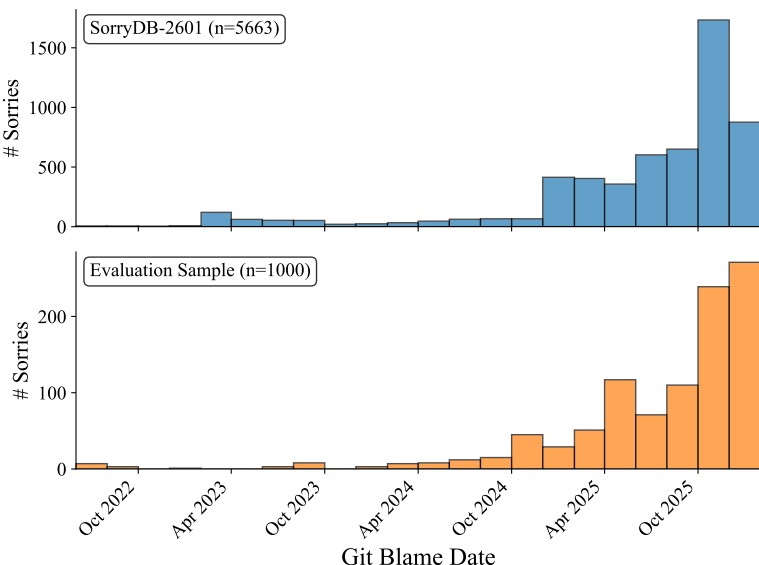

*Figure 6.* Distribution of git blame dates in SorryDB-2601 and the 1000 sorry sample. Serving as a proxy for the time of addition, these dates reveal that most `sorry` statements are very recent.

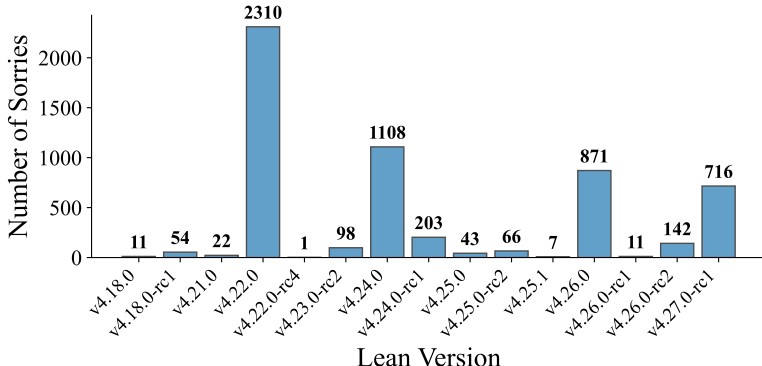

*Figure 7.* Lean versions of `SorryDB-2601` are distributed across major versions and release candidates with the vast majority of `sorry` statements on major versions 4.22, 4.24, 4.26, and 4.27.0-rc1, the last being the latest version at the time of the dataset collection.

## A.1. Total `sorry` statements on GitHub over time

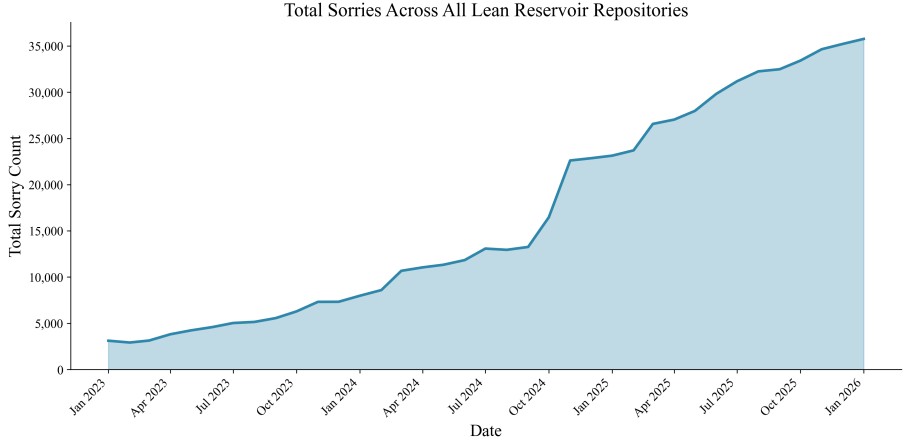

*Figure 8.* Total `sorry` statements on GitHub over time have steadily increased over the last 3 years.

## A.2. Dynamics of a Lean project

Figure 9 illustrates the dynamic nature of a representative repository, where periods of rapid development introduce many `sorry` statements, followed by gradual discharge as proofs mature.

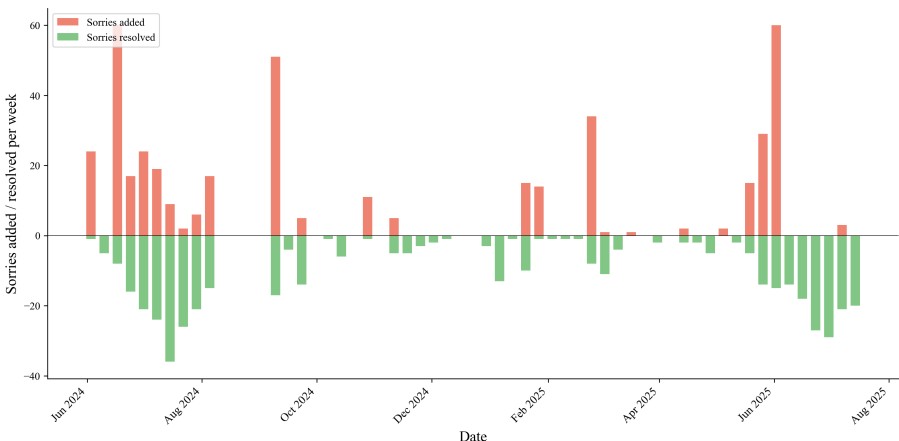

*Figure 9.* `sorry` statements added and removed from the Carleson project from its announcement to its completion. You can see sharp increases in number of `sorry` statements when new sections were added with missing proofs and then a gradual decrease as Lean practitioners provided proofs.

# B. Full dataset schema

We provide the full schema for the `SorryDB-2601` dataset.

*Table 3.* Schema of a Sorry record in the SorryDB database.

| Field | Type | Description |
|---|---|---|
| *repo* | | |
| remote | string | Repository URL |
| branch | string | Git branch name |
| commit | string | Git commit hash |
| lean_version | string | Lean version (e.g., v4.26.0) |
| *location* | | |
| path | string | File path within repository |
| start_line | int | Starting line number |
| start_column | int | Starting column number |
| end_line | int | Ending line number |
| end_column | int | Ending column number |
| *debug_info* | | |
| goal | string | Proof goal state at the sorry |
| url | string | URL to sorry location in repo |
| *metadata* | | |
| blame_email_hash | string | Hashed email of author |
| blame_date | datetime | Date sorry was introduced |
| inclusion_date | datetime | Date added to database |
| id | string | SHA256 hash identifier |

# C. Examples of `sorry` statements

We provide examples from the `SorryDB-2601` dataset.

---

**Example 1: Intermediate Value Theorem (RealAnalysisGame)**

| | |
|---|---|
| **Repository:** | AlexKontorovich/RealAnalysisGame |
| **File:** | Game/Levels/L25Levels/L02.lean:128 |
| **Lean Version:** | v4.23.0-rc2 |
| **URL:** | View on GitHub |

**Code Context:**

```
  /--
The Intermediate Value Theorem (IVT) states that if a function is continuous on a
    closed interval `[a, b]` and takes values `f(a) < 0` and `0 < f(b)`, then there
    exists `c ∈ (a, b)` so that `f(c)=0`.
-/
TheoremDoc RealAnalysisGame.IVT as "IVT" in "f(x)"

Statement IVT {f : ℝ → ℝ} (hf : FunCont f) {a b : ℝ} (hab : a < b)
    (hfa : f a < 0) (hfb : 0 < f b): ∃ c ∈ Ioo a b, f c = 0 := by
let S := { x ∈ Icc a b | f x < 0 }
have a_in_S : a ∈ S := by
  split_ands
  · bound
  ...
  linarith [hc.2 (c - δ / 2) cUB, hδpos]
  sorry
```

**Goal:**

---

```
  case refine_1
  f : ℝ → ℝ
  hf : FunCont f
  a b : ℝ
  hab : a < b
  hfa : f a < 0
  hfb : 0 < f b
  S : Set ℝ := {x | x ∈ Icc a b ∧ f x < 0}
  c : ℝ
  hc : IsLUB S c
  fc : f c = 0
  hca : a ≠ c
  ⊢ a < c
```

### Example 2: Field Multiplication (curve25519-dalek-lean-verify)

**Repository:**   Beneficial-AI-Foundation/curve25519-dalek-lean-verify
**File:**         Curve25519Dalek/.../FieldElement51/Mul.lean:44
**Lean Version:** v4.24.0
**URL:**          View on GitHub

**Code Context:**

```
/-
natural language description:

    · Computes the product of two field elements a and b in the field F_p where p =
    2^255 - 19
    · The field elements are represented as five u64 limbs each

natural language specs:

    · The function always succeeds (no panic)
    · Field51_as_Nat(result) ≡ Field51_as_Nat(lhs) * Field51_as_Nat(rhs) (mod p)
-/

/-- **Spec and proof concerning `backend.serial.u64.field.FieldElement51.Mul.mul`**:
- No panic (always returns successfully)
- The result, when converted to a natural number, is congruent to the product of the
    inputs modulo p
- Input bounds: each limb < 2^54
- Output bounds: each limb < 2^52
-/
@[progress]
theorem mul_spec (lhs rhs : Array U64 5#usize)
    (hlhs : ∀ i < 5, lhs[i]!.val < 2 ^ 54) (hrhs : ∀ i < 5, rhs[i]!.val < 2 ^ 54) :
    ∃ r, mul lhs rhs = ok r ∧
    Field51_as_Nat r ≡ (Field51_as_Nat lhs) * (Field51_as_Nat rhs) [MOD p] ∧
    (∀ i < 5, r[i]!.val < 2 ^ 52) := by
  sorry
```

**Goal:**

```
  lhs rhs : Array U64 5#usize
  hlhs : ∀ i < 5, ↑lhs[i]! < 2 ^ 54
  hrhs : ∀ i < 5, ↑rhs[i]! < 2 ^ 54
  ⊢ ∃ r, mul lhs rhs = ok r ∧
      Field51_as_Nat r ≡ Field51_as_Nat lhs * Field51_as_Nat rhs [MOD p] ∧
      ∀ i < 5, ↑r[i]! < 2 ^ 52
```

---

**Example 3: Division Algebra Finiteness (FLT)**

**Repository:**     ImperialCollegeLondon/FLT
**File:**           FLT/DivisionAlgebra/Finiteness.lean:106
**Lean Version:**   v4.27.0-rc1
**URL:**            View on GitHub

**Code Context:**

```
lemma existsE : ∃ E : Set (D_A), IsCompact E ∧
    ∀ φ : D_A ≃ₜ+ D_A, addEquivAddHaarChar φ = 1 → ∃ e₁ ∈ E, ∃ e₂ ∈ E,
    e₁ ≠ e₂ ∧ φ e₁ - φ e₂ ∈ Set.range (Algebra.TensorProduct.includeLeft : D →ₐ[K]
    D_A) := by
  --have := MeasureTheory.QuotientMeasureEqMeasurePreimage.haarMeasure_quotient
  sorry -- **TODO** prove that if A is a locally compact ab group and Gamma is a
    cocompact
  -- subgroup then there's some positive real M such that if U ⊆ A and μ(U)>M then
  -- U -> A/Gamma isn't injective.
```

**Goal:**

```
K : Type u_1
inst†⁶ : Field K
inst†⁵ : NumberField K
D : Type u_2
inst†⁴ : DivisionRing D
inst†³ : Algebra K D
inst†² : FiniteDimensional K D
⊢ ∃ E, IsCompact E ∧
    ∀ (φ : D_A ≃ᵖ+ D_A), addEquivAddHaarChar φ = 1 →
      ∃ e₁ ∈ E, ∃ e₂ ∈ E, e₁ ≠ e₂ ∧
        φ e₁ - φ e₂ ∈ Set.range ↑Algebra.TensorProduct.includeLeft
```

## D. Indexer Details

For each selected repository, the indexer extracts all the `sorry` keywords used inside the project. For each `sorry`, we store specific metadata, including the specific repo and commit branch, its URL, the commit date. All `sorry`s that are intentionally committed to a main branch are included alongside those found on any work-in-progress branch.

Subsequently, `sorry`s are filtered so that they can be reproduced in an evaluation environment. Specifically, for each `sorry`, we clone the associated repo, check out the specific branch and make sure that the Lean project successfully builds. We remove the tasks that fail to build. Next, the indexer uses the Lean REPL (The leanprover-community, 2023) interaction tool to extract the `sorry` from the Lean file and verify that it is "prop-valued", that is, the `sorry` requires a *proof* (i.e. a *term* of type `Prop`) rather than a term of some other type (e.g. `Nat` or `Real`). Indeed, filling a `sorry` for a non-`Prop` term is considered out of scope because one cannot automatically evaluate the correctness of the filled value: intuitively, this corresponds to providing a *definition* rather than a proof. The `sorry`s are then deduplicated, removing those that have exactly the same formal goal and keeping only the most recent occurrence. Deduplication is important as many `sorry` statements are repeated across multiple commits across a single repo.

## E. List of repositories in the `SorryDB-2601` dataset

*Table 4.* Repositories in the `SorryDB-2601` Dataset and number of `sorry`s per repository.

| Repository | Test Set | Full Set |
|---|---|---|
| AlexKontorovich/PrimeNumberTheoremAnd | 21 | 136 |
| AlexKontorovich/RealAnalysisGame | 21 | 55 |

*Continued on next page*

| Repository | Test Set | Full Set |
|---|---|---|
| Beneficial-AI-Foundation/curve25519-dalek-lean-verify | 21 | 86 |
| Bergschaf/lean-banach-tarski | 1 | 1 |
| Clap-lang/clap-lean | 13 | 13 |
| Deducteam/lean2dk | 1 | 1 |
| FormalizedFormalLogic/Foundation | 21 | 33 |
| FredRaj3/SemicircleLaw | 21 | 35 |
| GasStationManager/SafeVerify | 1 | 1 |
| HEPLean/PhysLean | 21 | 27 |
| ImperialCollegeLondon/FLT | 21 | 74 |
| Ivan-Sergeyev/seymour | 2 | 2 |
| JadAbouHawili/Raymond-Smullyan-KnightsAndKnaves | 6 | 6 |
| MichaelStollBayreuth/EulerProducts | 1 | 1 |
| NyxFoundation/tsl-formal-verification | 1 | 1 |
| PatrickMassot/GlimpseOfLean | 20 | 70 |
| Paul-Lez/PersistentDecomp | 20 | 42 |
| RemyDegenne/CLT | 5 | 5 |
| RemyDegenne/brownian-motion | 20 | 31 |
| RemyDegenne/lean-bandits | 13 | 13 |
| Timeroot/Lean-QuantumInfo | 20 | 137 |
| VCA-EPFL/graphiti | 20 | 49 |
| Verified-zkEVM/ArkLib | 20 | 431 |
| Verified-zkEVM/VCV-io | 20 | 94 |
| Verified-zkEVM/clean | 18 | 18 |
| WuProver/groebner_proj | 6 | 6 |
| YaelDillies/LeanAPAP | 20 | 41 |
| YaelDillies/LeanCamCombi | 20 | 45 |
| YaelDillies/MiscYD | 20 | 36 |
| YaelDillies/Toric | 9 | 9 |
| YellPika/quasi-borel-spaces | 2 | 2 |
| acmepjz/lean-iwasawa | 6 | 6 |
| alexkeizer/QpfTypes | 2 | 2 |
| apnelson1/Matroid | 20 | 61 |
| arademaker/fad | 20 | 22 |
| dagurtomas/LeanCondensed | 13 | 13 |
| djvelleman/HTPILeanPackage | 20 | 259 |
| dwrensha/compfiles | 20 | 120 |
| emilyriehl/infinity-cosmos | 1 | 1 |
| fpvandoorn/LeanCourse25 | 20 | 443 |
| fpvandoorn/carleson | 20 | 86 |
| frenzymath/jixia | 1 | 1 |
| frenzymath/reap | 1 | 1 |
| goens/lost-pop-lean | 10 | 10 |
| google-deepmind/formal-conjectures | 20 | 1698 |
| google-deepmind/formal-imo | 20 | 301 |
| jcreedcmu/Noperthedron | 20 | 29 |
| jsm28/IMOLean | 20 | 28 |
| kbuzzard/ClassFieldTheory | 20 | 55 |
| kebekus/ProjectVD | 17 | 17 |
| knowsys/Formale-Systeme-in-LEAN | 20 | 23 |
| lean-dojo/LeanMillenniumPrizeProblems | 14 | 14 |
| leanprover-community/aesop | 14 | 14 |

| Repository | Test Set | Full Set |
|---|---|---|
| leanprover-community/batteries | 7 | 7 |
| leanprover-community/duper | 2 | 2 |
| leanprover-community/mathlib4 | 20 | 21 |
| leanprover-community/quote4 | 1 | 1 |
| leanprover-community/sphere-eversion | 6 | 6 |
| leanprover/verso | 4 | 4 |
| m4lvin/Gossip-in-Lean | 19 | 19 |
| m4lvin/lean4-pdl | 20 | 47 |
| madvorak/chomsky | 1 | 1 |
| mo271/FormalBook | 20 | 78 |
| oliver-butterley/SpectralThm | 20 | 27 |
| or4nge19/NeuralNetworks | 5 | 5 |
| peabrainiac/lean-catdg | 14 | 14 |
| pitmonticone/ItaLean2025 | 16 | 16 |
| powdr-labs/leanr | 1 | 1 |
| rkirov/category-theory-in-context-lean | 21 | 31 |
| siddhartha-gadgil/LeanLangur | 10 | 10 |
| siddhartha-gadgil/LeanLion | 2 | 2 |
| siddhartha-gadgil/MetaExamples | 6 | 6 |
| sunblaze-ucb/verina | 8 | 8 |
| sven-manthe/A-formalization-of-Borel-determinacy-in-Lean | 4 | 4 |
| thefundamentaltheor3m/Sphere-Packing-Lean | 20 | 152 |
| ufmg-smite/lean-smt | 6 | 6 |
| vihdzp/combinatorial-games | 2 | 2 |
| yangky11/miniF2F-lean4 | 20 | 488 |
| **Total** | **1000** | **5663** |

## F. Verification infrastructure

LeanInteract queries the Lean REPL to report all `sorry` statements present in a file. We verify that (1) the modified file compiles successfully, (2) the number of `sorry` statements has decreased by exactly one, and (3) all remaining `sorry` statements have unchanged goals. Simply building the Lean project with lake build is insufficient because sorry is a valid Lean term that compiles without error. Our verification must therefore explicitly check that the proposed proof eliminates the sorry term rather than merely producing a syntactically valid file.

## G. Methodology for selecting the 1000 `sorry` statements test set

We extract `sorry` statements exclusively from leaf commits (branch tips). This ensures we capture incomplete proofs on the main branch as well as work-in-progress on individual development branches. For the evaluation, we selected 1000 `sorry` statements from SorryDB-2601 favoring both recency and diversity of repos: we selected the most recent sorry from each repo by blame date while pulling cyclically through repos. This prioritization of recency ensures the dataset reflects the current state of proof obligations and the real-world challenges Lean practitioners face using currently available tools.

# H. Tactic study

We conduct a study of the deterministic Tactic strategy used on the `SorryDB-2601` dataset to understand which individual tactics contribute most to its overall performance and how their coverage overlaps.

Figure 10 reports the total number of `sorry` statements solved by each tactic. General purpose automation tactics such as `grind` and `aesop` dominate along with the `exact?` library-search tactic.

Figure 11 breaks these results down by category using an UpSet plot, showing both unique and overlapping contributions of each tactic across repository categories. The tactic `exact?` uniquely resolves the largest number of `sorry` statements, which are predominantly drawn from pedagogical repositories whose exercises are often closed by a single Mathlib lemma. In contrast, `grind` solves `sorry` statements from a broader distribution of categories, demonstrating its versatility as a general-purpose tactic.

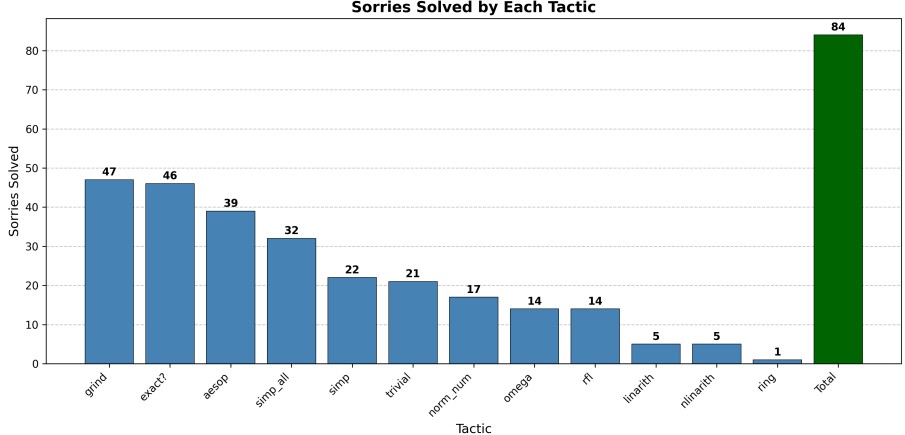

*Figure 10.* Number of `sorry` statements solved by each tactic. General purpose automation tactics like `grind` and `aesop` perform best.

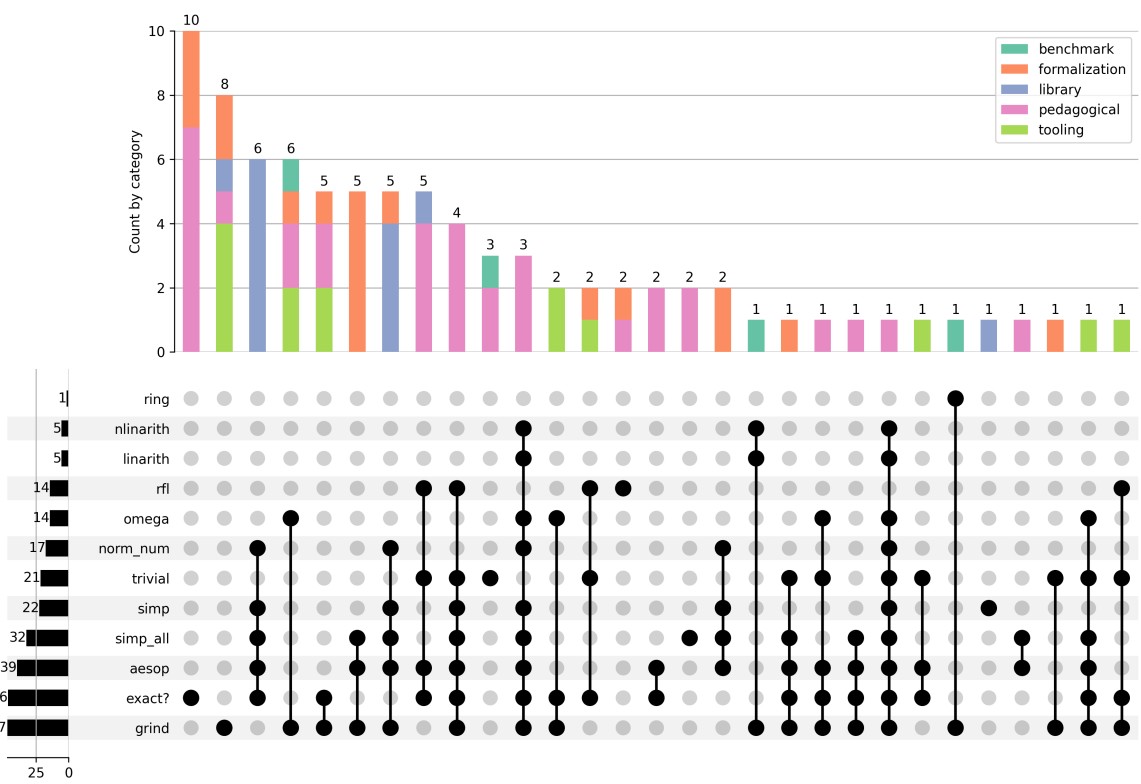

*Figure 11.* UpSet plot of `sorry` statements solved by each tactic. Interestingly, `exact?` uniquely solves the most `sorry` statements, mostly from pedagogical repos. This makes sense as tutorials are likely to have exercises easily dispatched by theorems already in Mathlib. Grind is in second place, but solves `sorry` statements from a wider distribution of categories showing its versatility.

# I. Analysis of token consumption and cost practicality

In Figure 12, we analyze the distribution of total tokens for each different model, considering only proposed proofs that were successful.

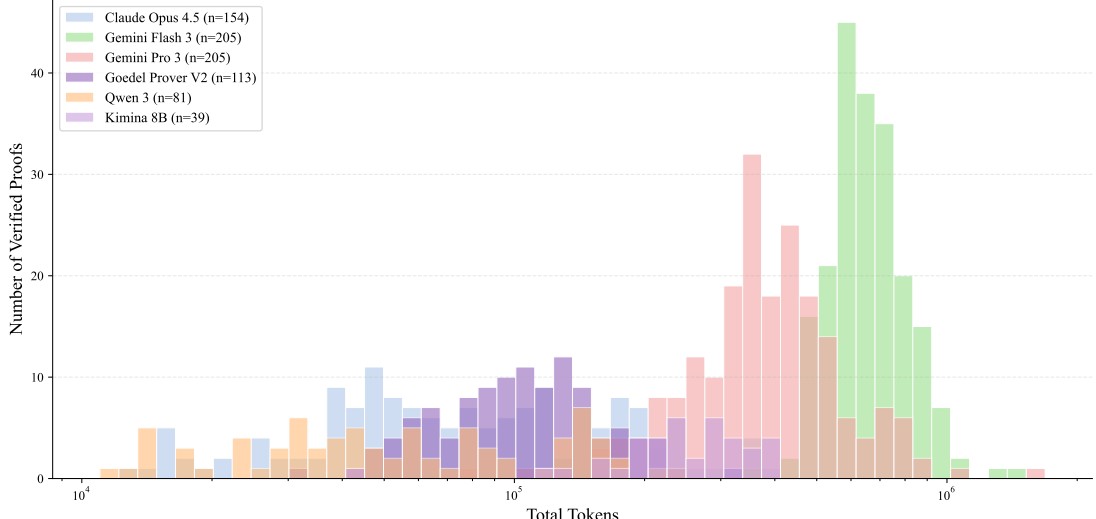

*Figure 12.* Token usage distributions for each model.

*Table 5.* Inference cost by strategy.

| Strategy | Total Input | Total Output | Est. Cost |
|---|---|---|---|
| Tactics | 0 | 0 | N/A |
| gpt-5.2 | 112,360,153 | 8,339,934 | $313.39 |
| Claude Opus 4.5 | 135,067,959 | 17,036,984 | $1,101.26 |
| Gemini Flash 3 | 118,964,320 | 643,259,636 | $1,989.26 |
| Gemini Pro 3 | 114,786,853 | 448,730,558 | $5,614.34 |
| Kimina 8B | 73,133,062 | 185,096,381 | $482.10[*] |
| Goedel Prover V2 | 79,339,958 | 67,794,892 | $1,227.36[*] |
| Qwen 3 | 108,441,104 | 6,850,768 | $171.23 |
| Claude Opus 4.5 (SC) | 114,848,662 | 52,675,745 | $1,891.14 |
| Gemini Flash 3 (SC) | 102,646,610 | 246,774,056 | $791.65 |
| Gemini Flash 3 (agentic) | 391,014,404 | 257,244,513 | $967.24 |

[*]Upper bound based on cost of self-hosting models

# J. Benchmark Governance

Here we outline the governance framework we adopt for SorryDB by proposing policies for versioning, reproducibility, and contamination control.

**Snapshot versioning and reproducibility.** Each release of SorryDB is a frozen snapshot identified by a `SorryDB-YYMM` tag (e.g., `SorryDB-2601` for January 2026). A snapshot pins every sorry to its exact repository, commit hash, branch, file path, and line range (see Table 3), making results fully reproducible. Snapshots are produced on a six-month cadence and once published are immutable.

**Contamination audit protocol.** Because SorryDB draws from public repositories, proof obligations may eventually enter model pretraining corpora. We propose three measures to limit contamination:

1. **Time-based cutoffs.** Each snapshot records the `blame_date` of every sorry and if needed researchers can filter based on training cutoff.

2. **Goal-state hashing.** The `goal` field captures the formal proof state at each sorry location. By hashing goal states across snapshots, we can flag `sorry` statements whose proof obligations appeared in earlier releases or in known training sets. While not a perfect novelty guarantee (the same goal can arise in different contexts), matching on goal-state hashes provides a practical, automatable signal for detecting potential leakage.

3. **Repository-level controls.** SorryDB maintains an explicit list of included repositories (Appendix E). This allowlist can be audited against known training corpora, and repositories suspected of contamination can be excluded from future snapshots.

## K. Elo Ratings

Elo ratings allow for long-term tracking of prover performance and provide a cross-snapshot comparability mechanism. We propose adopting an Elo rating system (Király & Qian, 2017), similar to CodeClash (Yang et al., 2025b). Under this scheme, provers accumulate a rolling rating based on head-to-head outcomes on shared or overlapping tasks, providing a relative performance measure that is meaningful across dataset versions. Concretely, we obtain the ratings reported here by fitting a Bradley-Terry model (Bradley & Terry, 1952) to the per-task win/loss outcomes between provers, with task ratings and prover ratings estimated jointly on the same logistic scale. This complements per-snapshot absolute metrics with a signal over time.

Figure 13 shows the distribution of task difficulty in the `SorryDB-2601` test sample, where each task is assigned an Elo rating derived from the head-to-head outcomes of the evaluated provers. Table 6 reports the resulting Elo ratings of the provers themselves, alongside their per-task accuracy on the same sample.

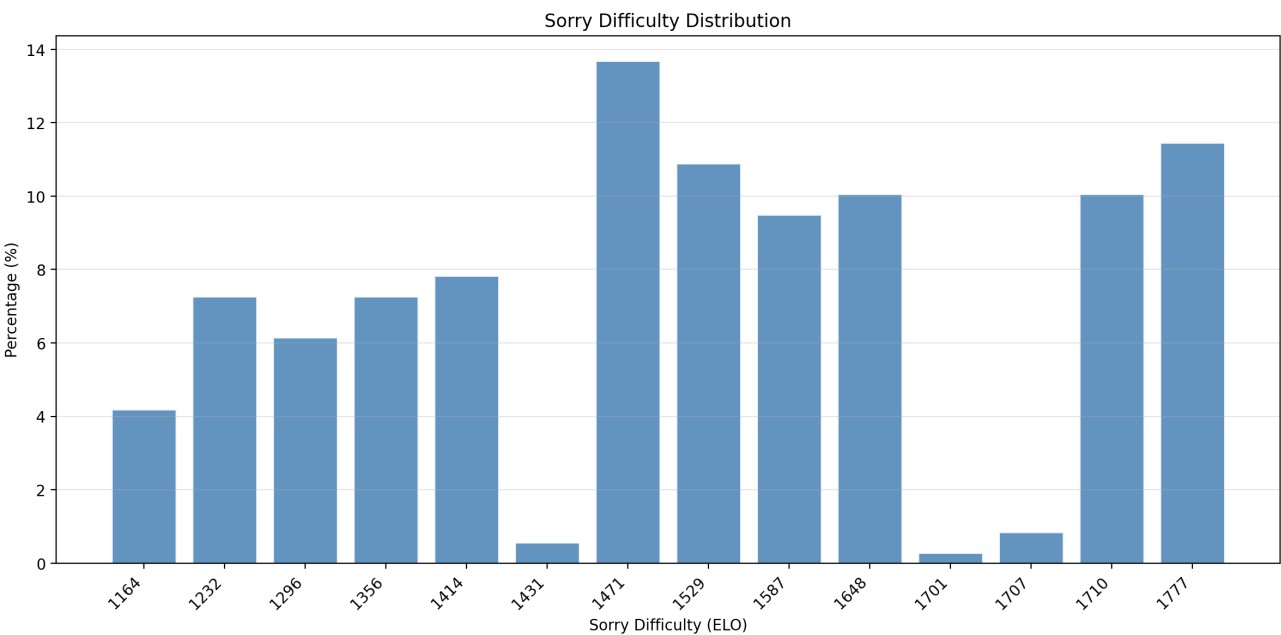

*Figure 13.* Distribution of `sorry` task difficulty in the `SorryDB-2601` test sample, expressed as Elo ratings derived from head-to-head prover outcomes.

*Table 6.* Elo ratings and accuracy of the evaluated provers on the `SorryDB-2601` test sample.

| Name | Accuracy | Elo |
|---|---|---|
| Gemini Flash 3 (Agentic) | 30.3% | 1859 |
| Gemini Flash 3 (SC) | 27.9% | 1775 |
| Claude Opus 4.5 (SC) | 27.1% | 1751 |
| Gemini Pro 3 | 20.6% | 1582 |
| Gemini Flash 3 | 20.5% | 1579 |
| Claude Opus 4.5 | 15.4% | 1457 |
| GPT 5.2 | 13.2% | 1403 |
| Goedel Prover V2 32B | 11.3% | 1354 |
| Tactics | 8.4% | 1270 |
| Qwen 3 | 8.1% | 1260 |
| Kimina Prover 8B | 6.6% | 1210 |

