# OpenReview forum: "SorryDB: Can AI Provers Complete Real-World Lean Theorems?"
_ICML.cc/2026/Conference — ICML 2026 regular_

### Official Review · Reviewer_G1Ek · 2026-03-10

**Soundness:** 3
**Presentation:** 3
**Significance:** 4
**Originality:** 3
**Overall Recommendation:** 5
**Confidence:** 4

**Summary:**

The study introduces a dynamic benchmark for formalization and theorem proving in Lean. The SorryDB dataset is based on actively maintained GitHub projects from the Lean Reservoir, with the recent snapshot containing 5663 problems. Data samples are diverse and directly extracted from open problems (focus on unsolved tasks, ”sorries”) in real-world projects and include libraries, formalization projects as well as competition problems. The authors position the dynamically updated benchmark as more applicable to real-world tasks and highlight its difficulty due as there typically no ground truth solutions available on from other web sources. The authors conduct experiments with open-source models as well as closed source API-based agents. The performance of the best performing models remains low, with higher success rates for simpler pedagogical problems. The authors find that open-source provers primarily trained and evaluated on competition problems fall short on real-world tasks included in SorryDB.

**Compliance With Llm Reviewing Policy:**

Affirmed.

**Final Justification:**

The authors propose a dynamic benchmark covering real-world open problems based on Lean Reservoir projects. Model evaluation and ranking on a dynamic benchmark containing problems with varying difficulty remains a concern. The authors suggest reasonable approaches addressing the issues. While evaluating formalization and prover performance on real-world math projects is not novel, the benchmark provides an incentive to evaluate on such problems, standardizes the format and removes erroneous examples. I believe my initial evaluation of the work is still appropriate.

**Key Questions For Authors:**

- How can the performance of different models be compared when the benchmark is updated in between evaluations?
- When a problem is solved and added to the github repo, the ground truth statement can be retrieved through web search. How do you deal with potential contamination?
- Do you plan on extending the project resources beyond the Lean Reservoir? If yes, how do you control for quality?
- Do you test whether the project and individual problems compile before integrating them in the benchmark?

**Limitations:**

yes

**Strengths And Weaknesses:**

## Strengths

- The experiments and discussion presented in the paper show interesting findings regarding agentic provers as well as open-source models, supporting the motivation that real-world tasks require additional resources in formal mathematics benchmarks.
- The benchmark is relevant as solving problems potentially benefits the community project the problem was extracted from.
- Good analysis of performance and limitations of current models and agentic approaches.

---

## Weaknesses

- There are several weaknesses and open questions that the authors discuss in Section 5. The benchmark in its current form does not allow a fair ranking of different models. The dynamic updates keep the problems relevant but make performance comparisons difficult.
- Evaluating formalization and prover performance on real-world math projects is not novel and has been explored in related studies. The authors acknowledge this and cite at least one such study.
- The problem difficulty varies widely across the different types of projects included in the benchmark. The current evaluation weights solved problems equally, which may distort the assessment of model capabilities.

---

> ### Author Rebuttal · Authors · 2026-03-31
>
> We thank the reviewer for the positive comments and suggestions. We respond to your questions and comments below:
>
> > How can the performance of different models be compared when the benchmark is updated in between evaluations?
>
> 1. As a benchmark, SorryDB is meant to evaluate the usefulness of a prover on real projects. For this reason, comparing models with outdated snapshots may be misleading, because older snapshots have higher probability of data contamination, and because strong performance on old projects may not imply immediate practical usefulness of the prover.
>
> In this paper, we report all results on the fixed snapshot of the dataset that we release, SorryDB-2601, which provides the basis for direct comparison. In the future, we plan to continue releasing new versioned snapshots (e.g. at 3 month intervals), so that comparison remains well defined within each snapshots, and with each new snapshot reflecting current practical utility. In addition, we intend to also provide a live evaluation infrastructure with a public leaderboard, similar in spirit to systems such as CodeClash.ai for SWE agents.
>
> Regarding weighting different difficulties, we agree that raw accuracy alone may not fully capture model strength when task difficulty varies substantially across tasks. As an additional analysis, we therefore consider a Bradley-Terry model, which assigns a relative score to both models and tasks according to which tasks each agent solves compared to other agents  and to how many models solve a given problem. This provides a principled way to report relative model strength and weight tasks according to relative difficulty. We report below our results:
> https://anonymous.4open.science/r/icml-sorrydb/ELO-models.md
> https://anonymous.4open.science/r/icml-sorrydb/ELO-problems.png
>
>
> >When a problem is solved and added to the github repo, the ground truth statement can be retrieved through web search. How do you deal with potential contamination?
>
> 2. We think contamination in the evaluations is unlikely, as the tasks we evaluate on are generated in a snapshot made at evaluation-time. Hence the tasks are unproven sorries in Lean projects. Moreover, the models we considered in our evaluation were not given web search access, so they could not retrieve solutions from the internet that might have been produced in the 2 week gap between snapshot creation and evaluation. While this does not fully eliminate the risk of contamination, it substantially reduces it in comparison to static benchmarks whose solutions are notoriously public. For broader community use of SorryDB, we believe contamination concerns can be greatly alleviated by evaluating on the most recently released snapshot, for which LLMs would be unlikely to have been trained on.
>
>
> >Do you plan on extending the project resources beyond the Lean Reservoir? If yes, how do you control for quality?
>
> 3. We currently plan to stay inside the Lean reservoir, as Reservoir already provides a useful quality filter, and it includes open source public projects that are selected according to strict community standards. This makes it the natural starting point for SorryDB, helping to ensure a consistent quality standard by construction. As a first step, we would first expand our coverage to a larger number of projects within the Reservoir. In the future, expanding beyond Reservoir may be interesting, but it would require introducing additional quality control criteria. We thought it best to stay within high-quality projects to ensure high quality of SorryDB.
>
> >Do you test whether the project and individual problems compile before integrating them in the benchmark?
>
> 4. Yes, before inclusion in the evaluation, each task is validated within its specific environment (Lean and Mathlib version at the specific commit), and those that do not build are excluded. This ensures compilation issues after applying a proposed proof are due to the proof itself and not to a pre-existing broken project.

---

> > ### Author Rebuttal · Reviewer_G1Ek · 2026-04-01
> >
> > Thank you for the detailed response to my comments. I believe my current evaluation of the work is still appropriate.

---

> > > ### Author Response · Authors · 2026-04-07
> > >
> > > Thank you for taking the time to review our work.

---

### Official Review · Reviewer_a3mu · 2026-03-12

**Soundness:** 4
**Presentation:** 2
**Significance:** 3
**Originality:** 3
**Overall Recommendation:** 4
**Confidence:** 4

**Summary:**

SorryDB is a benchmark in Lean 4 aimed to address the saturation and data-contamination issues in static competition math datasets. The benchmarks draws from 78 active open-source Lean formalisation projects on GitHub, continuously extracting incomplete proof steps marked by the `sorry` placeholder. The paper details the automated pipeline for scraping, filtering, and categorising the extracted tasks across different domains such as formalisation, pedagogical materials, libraries, and tooling. Furthermore, it presents an automated validation framework using LeanInteract to verify whether AI-generated proofs compile within their specific local repository contexts. Finally, the paper establishes a broad evaluation methodology, which combines testing deterministic tactics, foundation models (with multiple inference scaffolds), specialised provers at a static snapshot in January 2026.

**Compliance With Llm Reviewing Policy:**

Affirmed.

**Final Justification:**

I would like to thank the authors for the clarification and the valuable and insightful discussion.

Given the limitations clarified, I will keep my original positive score. To improve upon my assessment, I would have to see some of the limitations addressed in future work.

**Key Questions For Authors:**

1. The evaluation of the Agentic and Self-Correction (SC) harnesses is currently limited to Claude Opus 4.5 and Gemini Flash 3. Why were specialized models (e.g., Goedel Prover V2, Kimina Prover) or other foundation models not evaluated using these same harnesses? Could the authors provide a broader evaluation snapshot in the rebuttal, or elaborate on whether specialised provers inherently struggle to integrate with these specific iterative setups?

2. The paper currently lacks a discussion regarding how the local repository context is provided to the models. How is the relevant context extracted for each sorry task? Were the AI models able to access the repository-local definitions or intermediate lemmas defined within the same file? Additionally, for the agentic baseline, is the library search tool able to query the specific version of `mathlib` associated with that repository's Lean version?

3. The paper effectively highlights the dynamic nature and sheer scale of SorryDB compared to static datasets and other multi-repo benchmarks like RLMEval (which focuses on 6 research-level Blueprint projects) and FormalML (which focuses on ML theory subgoal completion). Beyond the number of repositories, could the authors provide a head-to-head comparison of the types of mathematical domains and the distribution of difficulty covered by SorryDB versus these existing benchmarks?

**Limitations:**

yes

**Strengths And Weaknesses:**

# Strengths

**Scaling a Dynamic Evaluation Infrastructure:**
While prior works (such as RLMEval, FormalML, and miniCTX) have explored using real-world Lean projects for evaluation, this paper significantly scales this work to 78 active GitHub repositories. The primary strength of this work lies in the infrastructure required to set up this dynamic framework, which successfully establishes a much broader, contamination-resistant, and realistic test of an AI agent's practical utility in collaborative formalisation tasks.

**Comprehensive Evaluation Methodology:** This paper evaluates a broad spectrum of models, as well as deterministic approaches, to provide a comprehensive snapshot of the current state of formal theorem proving in Lean. By benchmarking deterministic tactics (`grind` and `simp`), general foundation models (with and without self-correction), and specialised open-weight provers such as Kimina Prover and Goedel Prover, the paper provides a robust snapshot of the current state of the art for a wide breadth of domains in the Lean 4 eco-system.

**Presentation:** The figures in this paper are high-quality and greatly improve the clarity of the narrative. For example, the upset plots succinctly communicate the complementarity of different provers.

**Qualitative Analysis:** The qualitative breakdown of specific examples (e.g. in the `Brownian-motion` repository) provides direct explanation and intuition into the performance increase from self-correction approaches (e.g. hallucination of non-existent library lemmas may be corrected).

# Weaknesses

**Incomplete Evaluation Protocol:** Only Claude Opus 4.5 and Gemini Flash 3 were tested with the Self-Correction (SC) and Agentic harnesses. From a scientific perspective, it would be highly beneficial to see a less biased evaluation where specialised provers (like Goedel V2 or Kimina) and other foundation models are also equipped with these harnesses. This would provide the community with specific information regarding which models (and why) benefit most from the additional scaffolding to inform future research and deployments.

**Lacking Evaluation Context Discussion:** The paper lacks discussion on how exactly the relevant context is extracted for each task. It is unclear if the models are able to read the rest of the repository or file, and especially if the agentic implementation enhanced with a library search tool is able to search the version of `mathlib` associated with the task. In general, from related work in the area, it is clear that context is a crucial component for AI models to provide formally correct proofs, and it would be great to know that enough of the context is stored for each task in the benchmark.

**Writing Quality & Typos:** While the overall narrative and motivation are clear, the manuscript contains a significant number of typos and awkward grammatical constructions that impede readability (e.g., "The call the evaluation task" [Line 159], "being them designed" [Line 272], "essentially not informative because too simple" [Line 227] and "multiple deterministic tactic" [Line 229]). A thorough proofreading pass is necessary for the camera-ready version to improve the flow of the paper and correct the typos.

---

> ### Author Rebuttal · Authors · 2026-03-31
>
> We thank the reviewer for their thorough review, the positive feedback, and insightful comments, which we address below.
>
> > The evaluation of the Agentic and Self-Correction (SC) harnesses is currently limited to Claude Opus 4.5 and Gemini Flash 3. Why were specialized models (e.g., Goedel Prover V2, Kimina Prover) or other foundation models not evaluated using these same harnesses? Could the authors provide a broader evaluation snapshot in the rebuttal, or elaborate on whether specialised provers inherently struggle to integrate with these specific iterative setups?
>
> 1. The goal of our evaluation is to compare a set of representative baselines, show that SorryDB is not saturated, and that different provers solve different subsets of tasks. Methodologically, we first ran a broad set of provers in the pass@32 setting, and then we applied the self-correcting and agentic approaches on the strongest-performing ones, in order to measure how far iterative correction can push performance on our dataset.
> In this work, we prioritized a focused comparison rather than a full range evaluation of models and harnesses. Specialized provers were included to compare single-shot and pass@k performance, while the iterative experiments were designed to test how far stronger reasoning models can exploit compiler feedback and tool use. In particular, prior work on Goedel Prover V2 shows a modest gain when using self-correction (88.1% pass@32 to 90.4% self-correcting on MiniF2F), suggesting that an iterative approach may help, but wouldn’t change the overall conclusions of our comparison. Extending the evaluation to specialized provers may be a valuable direction for future work.
>
> > The paper currently lacks a discussion regarding how the local repository context is provided to the models. How is the relevant context extracted for each sorry task? Were the AI models able to access the repository-local definitions or intermediate lemmas defined within the same file? Additionally, for the agentic baseline, is the library search tool able to query the specific version of mathlib associated with that repository's Lean version?
>
> 2. For each task, we give the model the local context of the file up to the target sorry keyword. Definitions or intermediate lemmas defined within the same file are accessible to the models. This approach gives a meaningful local project context while avoiding the additional design choices required for exploring the full project across different files.
> We kept the context simple to ensure the paper's core contribution remains focused on the dataset and evaluation protocol, rather than on building a stronger agent. We agree that developing an agent capable of fully leveraging project-wide context is an exciting direction for future research, which we will highlight in the revised manuscript. Ultimately, by establishing these baselines, we hope to encourage broad community adoption of SorryDB, driving the development of more capable provers that excel on real-world tasks.
> For the agentic baseline, the library search tool is using a self-hosted version of the publicly available LeanSearch service, using Mathlib 4.19. Even if it is not matching each repository’s Mathlib version, our results suggest that it is still a useful tool.
>
> > The paper effectively highlights the dynamic nature and sheer scale of SorryDB compared to static datasets and other multi-repo benchmarks like RLMEval (which focuses on 6 research-level Blueprint projects) and FormalML (which focuses on ML theory subgoal completion). Beyond the number of repositories, could the authors provide a head-to-head comparison of the types of mathematical domains and the distribution of difficulty covered by SorryDB versus these existing benchmarks?
>
> 3. In the linked table, we compare SorryDB-2601 with other existing Lean datasets or multi-repo benchmarks in terms of overlap and the specific domain they cover. We see that SorryDB covers several parts of the other popular datasets, while also including additional topics such as logic and foundations. On the other hand, SorryDB does not share theorems with RLMEval, since the latter extracts already proven theorems from the selected repositories, not the incomplete ones.
> https://anonymous.4open.science/r/icml-sorrydb/dataset-overview.md
> Regarding task difficulty, we build a dataset that should mimic what the community is working on. The current results, with the strongest methods solving about 30% of the tasks, suggest that the benchmark provides a useful signal for model evaluation, being neither saturated nor intractable.
>
> We thank the reviewer for the careful attention in highlighting several typos and cumbersome grammatical constructions. We will take your revisions into account, and thoroughly proof-read for the following version.

---

> > ### Author Rebuttal · Reviewer_a3mu · 2026-04-03
> >
> > I thank the authors for the detailed and pointed response. I have one more question still remaining regarding response 2.
> >
> > In your response you mention that "the library search tool is using a self-hosted version of the publicly available LeanSearch service, using Mathlib 4.19"; does LeanSearch also allow searching Mathlib, or do you also index the repository being evaluated (e.g. the FLT project)? A potential significant limitation of this work stems from the fact that models are only able to read the definitions and lemmas in the same file as they are being evaluated, simply lacking the context of the overall infrastructure present in the repository to fully grasp the semantics of the `sorry` theorems extracted.

---

> > > ### Author Response · Authors · 2026-04-05
> > >
> > > Thank you for your follow up question regarding the context provided to the models in our evaluation.
> > >
> > > The LeanSearch implementation we use only indexes Mathlib and not the local repository. We agree this is a limitation of the current setup, especially for tasks that rely heavily on project-specific infrastructure and can be lifted in future work.
> > >
> > > Regarding local file context, we give all models the same context of the single file where the sorry statement occurs. Giving models access to wider project-level context in addition to the existing local context is a promising direction for future work.
> > >
> > > Our evaluation goal in this paper was not to build the strongest possible agent, but to provide a simple baseline and keep the focus on the dataset and evaluation protocol. In the future, more repository-aware agents can challenge the baseline agent we provided. We will clarify this explicitly in the paper and add it to the limitations section. Please let us know if you have any further questions. We are happy to discuss further.

---

### Official Review · Reviewer_f8Js · 2026-03-12

**Soundness:** 2
**Presentation:** 3
**Significance:** 3
**Originality:** 3
**Overall Recommendation:** 4
**Confidence:** 3

**Summary:**

This paper introduces SorryDB, a benchmark built from real-world Lean formalization projects on GitHub. Unlike most prior benchmarks, which mainly focus on isolated competition-style problems, SorryDB targets more realistic proof obligations arising in actual Lean development and is designed to be continuously updated over time. The paper evaluates a range of methods on this benchmark, including specialized theorem provers, general-purpose LLMs, and agent-based systems.

**Compliance With Llm Reviewing Policy:**

Affirmed.

**Final Justification:**

The concerns are partially addressed. I will keep my original score.

**Key Questions For Authors:**

Do you have more careful analysis of the distribution of the benchmark, and the failure mode of models?

**Limitations:**

Yes

**Strengths And Weaknesses:**

Strengths

Originality:
The paper introduces a benchmark centered on real-world Lean formalization projects rather than isolated competition-style problems. This is a relatively novel and well-motivated direction for theorem proving evaluation.

Presentation:
The paper is generally well written and easy to follow. The motivation for moving beyond traditional benchmark settings is clearly explained, and the construction of the benchmark, evaluation protocol, and empirical findings are presented in a fairly organized way.

Significance:
The benchmark targets an important gap in current evaluation. If the goal is to build systems that are genuinely useful for mathematical formalization, then handling repository-level proof obligations is an essential step beyond solving isolated benchmark problems. In this sense, the paper moves the evaluation of AI theorem provers toward a more realistic and practically relevant setting.

Weaknesses

Soundness:
The quality and interpretability of the benchmark remain somewhat unclear. The set of sorry tasks is likely highly heterogeneous: some may correspond to meaningful and challenging proof obligations, while others may be trivial placeholders, temporary omissions, or tasks that are not really solvable in the current local setting without broader refactoring. Because of this, it is not yet fully clear what strong performance on this benchmark actually demonstrates, or how representative the benchmark is of genuinely important formalization capabilities. The paper would be stronger if it provided a more careful analysis of benchmark quality, task validity, and the extent to which these sorry instances form a reliable evaluation set.

---

> ### Author Rebuttal · Authors · 2026-03-31
>
> We thank the reviewer for their positive assessment and for highlighting the importance of evaluating theorem provers on realistic, repository-level tasks. We appreciate your insightful questions regarding the benchmark’s task distribution, quality, and model failure modes, which we address below:
>
> > Do you have more careful analysis of the distribution of the benchmark,
> and the failure mode of models?
>
> Benchmark Quality and Task Distribution:
> We completely agree with the reviewer that proof obligations in real-world repositories are highly heterogeneous. We capture this reality in SorryDB because we believe this variety provides the right opportunity to evaluate the practical usefulness of a prover and to capture the tasks that are at the forefront of the formal methods community. Strong performance implies the understanding of the various parts of Mathlib, the compatibility with Lean versions that are currently used in Lean projects, the understanding of local context and of the Lean constructs, patterns and metaprogramming schemes that are used in different circumstances. This dataset is therefore complementary to the evaluations on competition math, which often emphasize reasoning skills, or on datasets on subfields of math, which focus on specific topics.
>
> Furthermore, the Lean repositories included in SorryDB are all sourced from the Reservoir package registry, where a baseline level of quality is required.
>
> In addition to the absolute score, we believe that the comparison among different approaches provides value: it is reasonable to expect that most provers will work equally well on trivial placeholders, and will also fail extremely hard tasks. The problems solved by a subset of the different approaches are the ones that provide the largest signal.
>
> The upset plot in Figure 4 shows the fraction of trivial tasks and how other tasks are distributed: over 1000 tasks, we see that 17 tasks can be solved by all approaches (are thus likely to be trivial) and 8 are solved by all except the tactic keywords. Moreover, the fact that the combined performance of all the approaches (35.7%) is larger than the performance of any single approach (max individual performance is 30.3% by Gemini Agentic), shows that provers are often complementary to each other. For practical purposes there may be value in trying one with a lower overall performance, if it solves a fraction of problems that is not solved by other provers.
>
> Failure Modes:
> Regarding the failure modes of the models, we provide [here](https://anonymous.4open.science/r/icml-sorrydb/failure-modes.png) an analysis of the most typical errors: tactic usage failures, remaining unsolved goals, other syntax errors, and attempts that contain additional sorry statements.
>
> For example, we see that Claude Opus tends to place sorry statements when uncertain, and that specialized models like Goedel and Kimina have a lower name error rate and syntax error rate compared to other models (even though fine-tuned on older Lean4 versions).

---

> > ### Author Rebuttal · Reviewer_f8Js · 2026-04-05
> >
> > The rebuttal partially addresses my concern by clarifying that the benchmark is intentionally heterogeneous and by providing some additional evidence that the dataset is not dominated by trivial tasks. However, my main concern is not only about triviality, but about task validity and interpretability: it remains unclear how many sorry instances correspond to meaningful standalone proof obligations, how many require broader refactoring or auxiliary lemmas, and what strong performance on this benchmark should concretely be taken to demonstrate.

---

> > > ### Author Response · Authors · 2026-04-07
> > >
> > > Thank you for encouraging a deeper discussion on this topic. We agree that the task validity and interpretation are crucial points for a benchmark.
> > >
> > > We first discuss task validity. Before including a task, we test the project successfully builds at the specific commit and that Lean REPL can successfully parse the goal state. In addition, we only include sorry statements which are prop-valued, that is, they are truly proof obligations and not stubbed out definitions which would be a less well defined task. Furthermore, at least 35.7% of the tasks in the evaluated 1000-task slice are demonstrably solvable under this protocol, since they were solved by at least one evaluated method.
> > >
> > > At the same time, we cannot have a reliable estimate of how the remaining unsolved tasks divide between solvable local obligations and tasks which are truly unsolvable. For heterogeneous Lean code, this distinction is difficult to determine automatically and often hard to infer even from manual inspection. We view this as a trade-off. By drawing tasks from active formalization projects rather than fully curating already proven statements, we cannot guarantee that every unsolved instance is self-contained. In return, the benchmark is grounded in the actual distribution of proof obligations that arise in current day-to-day Lean development, encouraging the development of theorem provers aligned with community needs. Strong performance on SorryDB demonstrates the ability to solve Lean tasks that the community is writing.
> > >
> > > More specifically, SorryDB measures practical proof-completion skills of theorem provers on real Lean projects, authored by the community. The analysis by category in Fig. 2 shows how performance on different categories is distributed and that different methods succeed on different tasks. In general, tasks in SorryDB capture a mix of relevant abilities, including good local context understanding, knowledge of Mathlib across the most popular versions that are currently used, mastering Lean syntax, and advanced mathematical reasoning skills. More expressive actions, like enabling broader refactoring of a task or the introduction of auxiliary lemmas are important features that should extend this submission in future work.

---

### Official Review · Reviewer_TsjK · 2026-03-13

**Soundness:** 3
**Presentation:** 3
**Significance:** 2
**Originality:** 3
**Overall Recommendation:** 4
**Confidence:** 4

**Summary:**

The paper presents SorryDB, an ever-evolving benchmark for automated theorem proving constructed by extracting unsolved problems from Lean repositories on GitHub. In total, the current snapshot of SorryDB contains 5663 unsolved problems.

Authors also evaluate several automated theorem proving approaches on a subset of 1000 problems from SorryDB. The evaluated methods include deterministic tactics, frontier models, agentic systems built on top of frontier models, Kimina 8B, and Goedel Prover V2 32B. Iterative agentic models dominate the benchmark with a maximum score of 30.3%, while the best specialized LLM prover only achieves 11.3% with pass@32.

**Compliance With Llm Reviewing Policy:**

Affirmed.

**Final Justification:**

Based on the author's rebuttal, particularly the clarification of the evaluation protocol, I'm raising my score to Weak accept.

**Key Questions For Authors:**

1. Can you please provide your measured performance of Kimina Prover and Goedel Prover V2 on MiniF2F? This would help clear up confusion about their very low measured performance on the "benchmarks" split of SorryDB.
2. Can you please clarify the following comment: [402r] "Finally, some solutions may potentially exploit Lean properties to trick the verification pipeline." Does this mean that the presented evaluation results cannot be fully trusted?
3. If you plan to release new snapshots of the benchmark in the future, what's your approach to ensuring that different theorem proving approaches will be comparable? The worry is that future publications might choose various different versions of SorryDB and the results might not be comparable.
4. Can you please state the protocol for choosing the version of Lean when evaluating on SorryDB? Should users use the latest Lean version, a fixed historic one, or a different one for each GitHub repository?

Answers to these questions will influence my scoring on the Soundness front.

**Limitations:**

yes

**Strengths And Weaknesses:**

*Note on notation*: By [104l] and [048r], I mean line 104 left and line 48 right, respectively.

**Soundness:**

Authors present a promising approach to evaluating existing automated theorem provers on SorryDB. However, some crucial details have to be addressed for the presented results to be sound:

- The very weak performance of Kimina-Prover-7B and Goedel-Prover-V2-32B, even on the "benchmarks" category (Figure 2), is suspicious and should be investigates. For context, Kimina-Prover-7B reports 63.1% pass@32 performance on MiniF2F, while Goedel-Prover-V2-32B reports 88.1% pass@32 on MiniF2F. To alleviate any suspicion about the evaluation protocol, I recommend including measured performance on yangky11/miniF2F-lean4, which is included in SorryDB.
- Related to the previous point: In Figure 5, the output of Goedel Prover contains numerous sorries, hinting that the evaluation protocol does not match the model's intended use. This deepens the suspicion that the model's low performance might have been causes by incorrect usage.
- [402r] "Finally, some solutions may potentially exploit Lean properties to trick the verification pipeline." - This is a serious limitation of the presented pipeline and should be mitigated before SorryDB can serve as a trustworthy benchmark.
- While the evolving nature of SorryDB has its benefits (discussed in Section 5), it necessitates a carefully considered protocol for releasing and labeling the individual snapshots. Authors should mention their proposed strategy for the frequency of updates and for the naming scheme. This is important because the main function of benchmarks is fair comparison of different methods, and in the future it'll be unclear on which snapshot to evaluate.
- The paper doesn't state which version of Lean was used during evaluation. Moreover, since different repositories use different Lean versions, choosing any one specific Lean version for evaluation might dramatically influence the benchmark's difficulty. The paper should address this problem and provide a clear evaluation protocol stating which Lean version(s) to use.

Other areas of improvement:
- The paper should state the total compute budget required for the evaluation of the different methods on the SorryDB subset of 1000 test samples, and also whether future papers should evaluate only on this test subset or on the whole dataset. If the compute budget required for evaluation on the whole benchmark is prohibitively expensive, it might negatively influence the usefulness of the benchmark.
- In the paragraph titled "Self-correcting and agentic approaches." in Section 4.2, the tested agentic approach should be described in more detail. In the current version, it's highly unclear how the agentic system looks like.
- [205r] "We describe our implementation in more detail in Appendix H." - Appendix H doesn't in fact describe the implementation in more detail. Therefore, it's unclear what the engineering challenge is and how it's solved. [199r] mentions "the evaluation requires cloning and building the specific project repository for each sorry" - it's unclear why the project has to be built for each sorry.
- [233r] "See Appendix H for a full analysis of the deterministic approaches." - No analysis of deterministic approaches is given in Appendix H (or any other appendix).
- [328l] "See the H appendix for more details of the successful deterministic tactic proofs." - Appendix H (or any other appendix) doesn't in fact contain details on successful deterministic tactic proofs.
- [372l] "Interestingly, the tool-enabled agentic approach failed to solve this same problem, as it kept calling the search tool rather than building the proof term itself." - This seems to indicate a problem in the system prompt of the agent, which should include more strict instructions for the search tool usage. The system prompt together with other details on the agentic approach should be included in the Appendix.
- [423l] "Trying to prove the negation of the goal for tasks without known solution could be a way to estimate this in the future." - This is incorrect, as some unprovability of a theorem does not imply provability of its negation.

**Presentation:**

The paper is presents most important details about the dataset and models evaluation in an understandable way.

Areas of improvement:
- It should be explicitly stated in early parts of the paper whether sorry tasks are extracted from all historical commits of a repository, or only from the most recent snapshot. This is only alluded to in Section 5 on the last page of the paper.
- [249l] "In particular, for practicality, we focus on a split of 1000 sorrys" - It should be explained why selecting only 1000 sorries is more practical. Is it because of limited compute?
- [1024] "The full methodology for curating the 1000 test" - This sentence seems to be incomplete.
- [220l] "A tactic is a short command or instruction that tells Lean how to construct a proof step-by-step, rather than you having to write out the complex logical terms yourself." - If an explanation of tactics and terms is to be given, the section should explain what "logical terms" are in this context.
- [299l] "... when working with SorryDB, access to the project environment and having the possibility to compile it to test a proposed solution is much more helpful than longer reasoning budget or fine-tuning on specific datasets." - The note about fine-tuning should be explained. It's unclear why the quantitative results reveal any information about the usefulness of fine-tuning, since none of the approaches were fine-tuned on SorryDB.
- In Figure 3, I would recommend using percentages on the y axis instead of absolute number of solved problems.
- [313l] "In addition to traditional competitive math benchmarks, we also include code generation benchmarks (Ye et al., 2025)." - Judged by the citation, the code-related benchmarks are not code-generation, but code-verification. This should be correctly stated, together with the specific names of the included benchmarks. Also, it should be clarified whether these additional benchmarks are included in SorryDB or only used in this specific experiment.
- [325l] "Finally, we notice that without library search provers struggle to find theorems on their own." - This should be explained in more detail. Which library search tools are used? Can their usefulness be somehow measured?
- In Figures 2 and 3, I recommend including which size variant of Goedel Prover V2 was used (i.e. "32B" according to Table 2).
- The case study of RemyDegenne/CLT should be accompanied by proof examples, similarly to examples in Figure 5 for the Brownian-motion repository. All of these examples can be in Appendix instead of the main text, as they are not immediately necessary for understanding.

Minor points:
- [257r] superfluous space before the period
- [272l] superfluous "them"
- [229r] missing comma
- [327l] missing paragraph end
- Figure 4 top: some legend items are overlayed by horizontal lines
- Figures 4 and 11 together with [077l]: "UpSet" instead of "Upset"

**Significance:**

While the concept of an ever-evolving benchmark of frontier formalization is useful, the current state of SorryDB does not enable trustworthy evaluation due to missing crucial details on the evaluation protocol and also due to technical problems.

However, I believe the concept of SorryDB has meaningful potential and I encourage authors to pursue this direction further.

**Originality:**

The paper presents a novel, useful approach to evaluating automated theorem provers on the frontier of formal theorem proving on public GitHub repositories.

Section 2 positions the paper in the context of existing work. I believe that LEAN-GitHub should be cited, as it also concerns extracting Lean theorems from GitHub.

---

> ### Author Rebuttal · Authors · 2026-03-31
>
> We thank the reviewer for their thoroughness and extremely thoughtful comments.
>
> 1. Our benchmark contains 20 sorries from miniF2F, Lean v4.24, on which we obtain a pass@32 performance of 12/20 for Kimina and 11/20 for Goedel. Following the reviewer’s suggestion, we also run both Goedel and Kimina, pass@32, on the full miniF2F with Lean version 4.9 for direct comparison with their papers.
> We get Kimina: 74% (close to their 78% https://huggingface.co/blog/AI-MO/kimina-prover) and Goedel: 75% (below their 88%) We then audited our Goedel’s configuration and identified the issue: we capped generation length at 3k tokens to mitigate instability observed during our testing on SorryDB (model entered in loops repeating the same sentence over and over when context was too long). This seems to negatively affect performance on miniF2F. With the context length suggested in their paper (24k), we get 85% performance on miniF2F v4.9. For Kimina, we additionally found that a cloud-provider error affected approximately 2% of tasks. These controls reduce the likelihood that the low score on SorryDB is caused by a basic implementation bug. The remaining performance gap on the miniF2F tasks on SorryDB can be explained by the different Lean4 version and low statistics, while we still see a more substantial gap between performance of Kimina and Goedel models on competition math and the overall SorryDB dataset. Updated values will be integrated in the manuscript.
>
> 2. We appreciate the vigilance regarding the automated verification and we agree that the manuscript should state the verifier’s guarantees more clearly. The risks we mentioned were theoretical risks that we specifically mitigate with our verification pipeline. Specifically, all solutions are structurally limited to only replacing the sorry statements, which prevents adding exploits to the surrounding code. Moreover, Lean Interact checks that the solutions compile and are sorry-free, string matching catches known exploits using metaprogramming, together complemented with manual inspection of the solution, which did not report any evidence of metaprogramming hacking. We therefore do not believe that the reported results are invalidated by verifier failure.
> However, we agree that automated soundness in the verification pipeline will induce more confidence in SorryDB and in other benchmarks. Existing community tools (e.g. Lean Comparator) are not directly applicable to our setting because we allow dependence on sorries in other theorems in the project. Therefore, we have developed a prototype kernel checking tool to strengthen this part of the pipeline in the future.
>
> 3. All reported results are on the frozen snapshot SorryDB-2601, which provides a basis for direct comparison. Since SorryDB is meant to evaluate usefulness on real projects, we believe comparisons should be made on the most recent available dataset snapshot. In future releases, we plan to publish additional named snapshots at regular intervals so that evaluations remain both comparable within a snapshot and representative of current practical utility.
>
> 4. SorryDB is not tied to a single Lean4 version. Each task is evaluated in its own repository environment, using the specific commit and Mathlib version associated with that task. Fig. 7 shows Lean version distribution in our dataset.
>
> Evaluation costs
>
> We consider 1000 problems, while PutnamBench is 672 and miniF2F is 244. We show the token usage and cost of our evaluation in this table: https://anonymous.4open.science/r/icml-sorrydb/compute-cost.md
>
> In addition, we perform a statistical analysis of the performance, showing how the ranking would degrade as we reduce the size of the dataset to 500 or 250 problems: (bootstrap on sets of N problems) https://anonymous.4open.science/r/icml-sorrydb/dataset-size-comparison.png
>
> [205r][199r] We expand Appendix F (mislabelled H): evaluating tasks across repositories and Lean versions is complex and each project can take over an hour to build and exceed 5GB in disk space.
> To address these challenges we use a distributed pipeline that first prepares and builds a virtual snapshot associated with each commit and then allows proposing a proof and verifying it in the associated project environment for multiple tasks in parallel.
>
> [233r][238l]: Appendix H contains figures about deterministic approaches (i.e. tactics, see Fig. 10 and 11). We will expand.
>
> [372l] The performance of the tool-enabled agent compared to the agent without tools shows that tools do not degrade performance (30% vs 27%). We rephrase the sentence and add details.
>
> Finally, we thank the reviewer for the thorough comments on the presentation, including mentioning earlier how sorry tasks are extracted, adding missing technical explanations on specialized (i.e. fine-tuned) models (Goedel and Kimina), additional qualitative examples in the appendix and proof reading among others. They will be addressed in the camera ready version of the manuscript.

---

> > ### Author Rebuttal · Reviewer_TsjK · 2026-04-02
> >
> > Thank you for your detailed rebuttal and for your contribution.
> >
> > I'm still concerned about the evaluation practicality on SorryDB, mainly because the benchmark is continually evolving through time, and it is not clear how to compare different methods evaluated at different times. My concern about missing evaluation protocol was not addressed. Since the benchmark is the main contribution of the paper, it should be clear to the reader how to evaluate a new method and how to compare the resulting score to previous existing ones.

---

> > > ### Author Response · Authors · 2026-04-06
> > >
> > > Thank you for your follow-up question regarding evaluation practicality. We entirely agree that a clear evaluation protocol is essential. We have structured our response into three parts: comparing future evaluations to the results in the paper, comparing results between different methods on different snapshots, and technical details of the evaluation protocol.
> > >
> > > The dataset used in this paper, `SorryDB-2601`, is a permanent, frozen snapshot that we make publicly available. Future researchers should evaluate their models on `SorryDB-2601` to report directly comparable raw accuracy against the baselines presented in this work.
> > >
> > > In the future, to maintain our goal of evaluating on the uncontaminated frontier of active formalization, we will regularly publish a similar snapshot using the same selection criteria. Our current plan is to release a new snapshot every six months. Snapshots will be named `SorryDB-YYMM`, e.g. the next snapshot will be `SorryDB-2607` generated at the beginning of July. When a new snapshot becomes available, we recommend using that one for new evaluations, as it will be most representative of the current distribution of proof obligations in real, ongoing formalization projects.
> > >
> > > Directly comparing raw accuracy results between different snapshots is currently not encouraged, because the field evolves so rapidly that it may not be informative enough, and the overlap between different snapshots is typically small. However, to support SorryDB’s vision as a continuously updating benchmark, we propose using scores obtained from a Bradley-Terry model (Elo-like score) as a supplementary measure, in addition to absolute accuracy. With the appropriate fixed reference provers across releases, this will enable comparison between different snapshots of SorryDB. While direct comparisons of accuracy should remain within a fixed snapshot, we believe it is useful to provide a principled supplementary mechanism for relating results across snapshots. We initially only mentioned this in the discussion section of the paper to avoid shifting focus from the dataset we introduced, however after discussion with the reviewers, we have decided to add an appendix which demonstrates this measure more thoroughly.
> > > We have attached the calculated Elo scores for the results in this paper:
> > > - [Model Elo scores](https://anonymous.4open.science/r/icml-sorrydb/ELO-models.md)
> > > - [Sorry task Elo scores](https://anonymous.4open.science/r/icml-sorrydb/ELO-problems.png)
> > >
> > > Regarding the concrete details of the evaluation protocol, we provide the code to run your own method on SorryDB either locally on your machine or using our parallelized remote infrastructure. The infrastructure we open source makes it straightforward to evaluate methods on SorryDB. In addition to the detailed instructions on the open-sourced repository, we will add an appendix to the camera ready version which outlines how to evaluate your own model on `SorryDB-2601` and future snapshots. Concretely, the user provides their own implementation of a `prove_sorry(sorry: Sorry)-> Proof` method that takes the specific context of a task as input and should return the code for the proposed proof.

---

### Decision · Program_Chairs · 2026-04-30

**Decision:**

Accept (regular)

**Comment:**

This paper introduces SorryDB, a highly valuable benchmark that extracts real-world proof obligations from active Lean repositories to address the critical issue of test-set contamination in AI theorem proving. Reviewers unanimously appreciated the significant engineering effort and the timely shift from evaluating on isolated competition math to practical, repository-level formalization tasks. During the rebuttal, the authors effectively addressed reviewers' concerns regarding dynamic evaluation fairness and baseline configurations by establishing fixed dataset snapshots and correcting context-length truncation bugs. Given its strong motivation, technical soundness, and substantial utility to the AI-for-math community, I recommend this paper for acceptance.